# The Brewer-Dobson circulation in CMIP6

Marta Abalos[1], Natalia Calvo[1], Samuel Benito-Barca[1], Hella Garny[2], Steven C. Hardiman[3], Pu Lin[4,5], Martin B. Andrews[3], Neal Butchart[3], Rolando Garcia[6], Clara Orbe[7], David Saint-Martin[8], Shingo Watanabe[9], and Kohei Yoshida[10]

[1]Universidad Complutense de Madrid, Madrid, Spain
[2]Deutsches Zentrum für Luft- und Raumfahrt (DLR), Oberpfaffenhofen, Germany
[3]Met Office Hadley Centre, Exeter, United Kingdom
[4]NOAA/Geophysical Fluid Dynamics Laboratory, Princeton, NJ, USA
[5]Program in Atmospheric and Oceanic Sciences, Princeton University, Princeton, NJ, USA
[6]National Center for Atmospheric Research, Boulder, CO, USA
[7]NASA Goddard Institute for Space Studies, New York, NY, USA
[8]Centre National de Recherches Météorologiques, Toulouse, France
[9]Japan Agency for Marine-Earth Science and Technology (JAMSTEC), Yokohama, Japan
[10]Meteorological Research Institute, Tsukuba, Japan

**Correspondence:** Marta Abalos (mabalosa@ucm.es)

**Abstract.** The Brewer-Dobson circulation (BDC) is a key feature of the stratosphere that models need to accurately represent in order to simulate surface climate variability and change adequately. For the first time, the Climate Model Intercomparison Project includes in its phase 6 (CMIP6) a set of diagnostics that allow for careful evaluation of the BDC. Here, the BDC is evaluated against observations and reanalyses using historical simulations. CMIP6 results confirm the well-known inconsistency in the sign of BDC trends between observations and models in the middle and upper stratosphere. Nevertheless, the large uncertainty in the observational trend estimates opens the door to compatibility. In particular, when accounting for the limited sampling of the observations, model and observational trend errorbars overlap in 40% of the simulations with available output. The increasing $CO_2$ simulations feature an acceleration of the BDC but reveal a large spread in the middle to upper stratospheric trends, possibly related to the parameterized gravity wave forcing. The very close connection between the shallow branch of the residual circulation and surface temperature is highlighted, which is absent in the deep branch. The trends in mean age of air are shown to be more robust throughout the stratosphere than those in the residual circulation.

## 1 Introduction

The Brewer-Dobson circulation (BDC) describes the net transport of mass, heat and tracers in the stratosphere, and therefore plays a primary role in its chemical composition and radiative transfer properties (Butchart, 2014). In particular, the strength of the BDC controls key features such as the rate of stratospheric ozone recovery (Karpechko et al., 2018), the

stratosphere-to-troposphere exchange of ozone (e.g. Albers et al., 2018), and the amount of water vapor entering the strato-sphere (Randel and Park, 2019). The BDC is also fundamentally connected with the thermal structure of the stratosphere and in particular the static stability around the tropopause (Birner, 2010), a key radiative forcing region that also influences deep

convection (e.g. Emanuel et al., 2013). Therefore, realistically representing the BDC strength and its variability is a key target for climate models.

The BDC is commonly separated into two components: the residual circulation, which is the mean meridional mass circu-lation approximating the zonal-mean Lagrangian transport, and two-way mixing, the irreversible tracer transport caused by stirring of air masses following wave dissipation (Plumb, 2002). The residual circulation in turn is typically divided into the

shallow and the deep branch, with the former approximately limited to latitudes below 50°N/S and levels below 50 hPa and overturning timescales under 1 year (Birner and Bönisch, 2011). In the literature, the term BDC sometimes refers to the resid-ual circulation alone, and previous multi-model assessments of the BDC focused on this component (i.e. Butchart et al., 2010; Hardiman et al., 2014). However, there is growing evidence over the last years of the important role of mixing for net strato-spheric tracer transport (Garny et al., 2014; Dietmüller et al., 2017; Eichinger et al., 2018; Ploeger et al., 2015). The mean age

of air (AoA) transport diagnostic quantifies the elapsed time since an air parcel entered the stratosphere, and it can be estimated from observations of long-lived tracers such as $SF_6$ or $CO_2$ (e.g. Engel et al., 2017). Therefore, it integrates the effect of both residual circulation and mixing. While there are no direct measurements of the BDC strength, model results can be evaluated against observational estimates of the AoA.

In this study we assess the climatology and trends in the BDC in CMIP6 models, with a focus on current open questions.

A key open question is the disagreement between observations and models regarding the past BDC trends. While models consistently predict a reduction in AoA mainly due to an acceleration of the residual circulation (Li et al., 2018), the longest observational estimates produce non-significant positive trends (Engel et al., 2017). Various studies over the last years suggest that an acceleration of the BDC might actually be observed in the lower stratosphere (see Karpechko et al. (2018) and references therein). Moreover, it has been argued that the formation of the ozone hole has had a significant impact on the past BDC

acceleration until the end of the 20th century (Oman et al., 2009; Li et al., 2018; Polvani et al., 2018; Abalos et al., 2019). However, the observation-model discrepancy remains at higher altitudes. Indeed, the trends in the deep branch and its drivers remain more uncertain (WMO, 2018), given the limitations of model top and the importance of parameterized gravity waves in the upper stratosphere and mesosphere. On the other hand, recent work has highlighted that the observational AoA trend estimates are likely biased high (Fritsch et al., 2020).

CMIP6 is the first CMIP activity providing Transformed Eulerian Mean (TEM) diagnostics and mean age of air (AoA) as model output (Gerber and Manzini, 2016). This allows for a more detailed analysis based on a consistent set of diagnostics as compared to previous assessments (e.g. Manzini et al., 2014; Hardiman et al., 2014). Here, we use this TEM and AoA output to evaluate the past climatology and trends in the BDC against reanalyses and observations using historical simulations, and to assess the BDC response to an idealized 1%/year $CO_2$ increase. In Section 2 we describe the CMIP6 models and simulations

used, as well as other datasets employed. Section 3 analyzes the BDC in historical simulations, Section 4 examines the BDC

**Table 1.** List of CMIP6 models with TEM and/or AoA diagnostics used in this study. AoA is only provided in the historical runs for MRI.

| Model | Model and data references | Levels | Model top | TEM | AoA |
|---|---|---|---|---|---|
| CNRM-ESM2-1 | Séférian et al. (2019); Seferian (2018a, b) | 91 | 78.4 km ($\sim$ 0.01 hPa) | ✗ | ✓ |
| CESM2-WACCM | Gettelman et al. (2019); Danabasoglu (2019a, b) | 70 | 0.0000045 hPa | ✓ | ✗ |
| MIROC6 | Tatebe et al. (2019); Tatebe and Watanabe (2018a, b) | 81 | 0.004 hPa | ✓ | ✗ |
| GFDL-ESM4 | Dunne et al. (2020); Krasting et al. (2018a, b) | 49 | 0.01 hPa | ✓ | ✓ |
| GISS-E2-2-G | Rind et al. (2020); Orbe et al. (2020); NASA/GISS (2019) | 102 | 0.002 hPa | ✓ | ✓ |
| UKESM-1-0-LL | Sellar et al. (2019); Tang et al. (2019a, b) | 85 | 85 km ($\sim$ 0.005 hPa) | ✓ | ✓ |
| HadGEM3-GC31-LL | Williams et al. (2018); Ridley et al. (2019a, b) | 85 | 85 km ($\sim$ 0.005 hPa) | ✓ | ✗ |
| MRI-ESM2-0 | Yukimoto et al. (2019a, b, c) | 80 | 0.01 hPa | ✓ | ✓ (historical) |

changes associated with increases in $CO_2$ and Section 5 explores the connections between BDC and surface warming. The main conclusions are summarized in the last Section.

## 2 Data and Methods

CMIP6 models providing the necessary TEM and/or AoA output, as described in Gerber and Manzini (2016), have been used. We refer to that paper for the specific diagnostic description. Table 1 shows the models used, their number of levels, the model top and the corresponding available variables. While each model has a different horizontal resolution (not shown), they are all in the range between 1 and 2 degrees in longitude and latitude. Note that two more models output TEM variables, CESM2 and Can-ESM5, but we did not include them in the analyses because they have low model tops (2.25 hPa and 1 hPa, respectively), and did not represent the residual circulation structure adequately (not shown).

Model results have been compared to reanalysis over the historical period. Because there is a large spread in the residual circulation values obtained from reanalyses (Abalos et al., 2015), we used three reanalyses: ERA-Interim (Interim European Centre for Medium-Range Weather forecasts Reanalysis, Dee et al. (2011)), JRA-55 (Japanese 55 year Reanalysis, Kobayashi et al. (2015)), and MERRA (Modern-Era Retrospective Analysis for Research and Applications, Rienecker et al. (2011)). We have used the residual circulation for these three reanalyses from Abalos et al. (2015), which covers the period 1979-2012. In order to compare AoA with observational estimates, we used AoA derived from the Michelson Interferometer MIPAS (Stiller et al., 2012; Haenel et al., 2015). Here, we use a new version of AoA derived from an updated retrieval of $SF_6$ (Stiller et al., 2020). Furthermore, AoA derived from GOZCARDS $N_2O$ data is used (Linz et al., 2017), as well as AoA data derived from in-situ measurements of $CO_2$ and $SF_6$ by Engel et al. (2017) and Andrews et al. (2001).

The model simulations used are historical and 1pctCO2. We use one member of each simulation because there is a very uneven number of members for the different models (from 1 to 18), and therefore the comparison across models would be

unfair if the ensemble mean were used for each model. Nevertheless, we do exploit the multiple members when available in order to explore the role of internal variability on the trends (Figures 5, 6, 7 and 10). The fully-coupled (DECK) historical simulations cover the period 1850 to 2014, with observed emissions of greenhouse gases and other external forcings, and will

be used to examine past climatology and trends and compare to observations or reanalysis when possible. For comparison purposes we have focused on the period 1975-2014. Note that this period encompassess the reanalysis period considered (1979-2012), but it is slightly longer. This is done to enhance statistical significance in trend calculations for model output. The small difference in the period considered is assumed to have a negligible impact on the climatological BDC. The 1pctCO2 simulations are initialized from preindustrial (1850) conditions and are 150 years long, with $CO_2$ concentrations increasing

gradually at a 1%/year rate (Eyring et al., 2016). In order to establish statistical significance we have used a two-tailed Student t test at the 95% confidence level. We further ask for two thirds (2/3) of the models to agree (that is, 5 out of 7 for the residual circulation and 4 out of 5 for AoA).

## 3  Representation of the BDC in CMIP6 historical simulations

This section aims to assess the degree of agreement in the BDC climatology and trends between CMIP6 historical simulations

and observations or reanalysis data.

### 3.1  Climatology and seasonality

Figure 1 shows the climatological structure of the residual circulation in the CMIP6 multimodel mean (MMM, panel b) compared with the multi-reanalysis mean (panel a). The climatological structure and magnitude are overall very similar in both datasets. Both models and reanalyses highlight a minimum in tropical upwelling at $\sim 50$ hPa of about 0.2 $mm \cdot s^{-1}$, and a

maximum at $\sim 1.5$–2 hPa of about 1.2 $mm \cdot s^{-1}$. The annual mean residual circulation structure in Fig. 1 is consistent with previous model intercomparison studies such as CMIP5 (Hardiman et al., 2014).

In order to examine the quantitative differences in more detail, the tropical upwelling mass flux is examined. This is computed as the net upwelling between the annual mean turnaround latitudes (i.e., the latitudes separating the upwelling and downwelling regions). The calculation is based on the streamfunction, which in turn is computed from the meridional component of the

residual circulation provided as model output, $\bar{v}^*$. The streamfunction is obtained as

$$\bar{\Psi}^*(\phi, p) = -\frac{cos\phi}{g} \int\limits_p^0 \bar{v}^* dp' \tag{1}$$

where $p$ is pressure, $\phi$ is latitude, $g$ the gravitational constant on Earth, and it is assumed that $\bar{v}^*$ tends to zero as $p \to 0$. The upwelling mass flux is then computed at each level as

$$M(p) = 2\pi a \left(\bar{\Psi}^*_{max}(p) - \bar{\Psi}^*_{min}(p)\right) \tag{2}$$

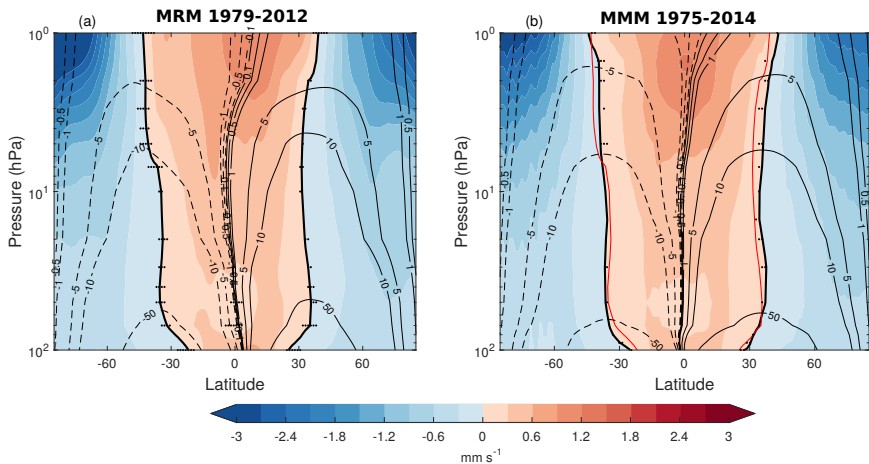

**Figure 1.** Annual mean climatology for the multi reanalysis mean (a) and multi model mean for the historical simulations (b) of the vertical component of the residual circulation ($\bar{w}^*$, in $mm \cdot s^{-1}$, shading) and residual streamfunction ($\Psi^*$, in $kg \cdot m^{-1} \cdot s^{-1}$, contours). Black thick contours indicate the location of the turnaround latitudes. The red contour in the right panel shows the turnaround latitudes for reanalyses. Black dots represent regions where there is disagreement in the sign of $\bar{w}^*$ for more than 66% (2/3) of the individual reanalyses or models (which happens only around turnaround latitudes).

where $\bar{\Psi}^*_{max}$ and $\bar{\Psi}^*_{min}$ are the maximum and minimum values of the residual streamfunction at each pressure level, which correspond to the northern and southern turnaround latitudes, respectively (Rosenlof, 1995).

Figure 2 shows the seasonality in the tropical upwelling mass flux for the lower (70 hPa) and upper (1.5 hPa) stratosphere, representative of the shallow and deep branches, respectively. Note that, while the level of 70 hPa is commonly used to represent the shallow branch, 1.5 hPa is higher than usually considered for the deep branch, though it has been used before

(Palmeiro et al., 2014). We argue that this level is optimal for the characterization of the deep branch, since tropical upwelling maximizes at this level in the upper stratosphere (Fig. 1). All models show a generally consistent seasonality, with an annual cycle peaking in November-December in the shallow branch, and an amplitude of about 50% of the climatological mean, and a semi-annual cycle peaking in June and December for the deep branch, with an amplitude of about 80%. The seasonality is consistent with that of reanalyses, and the intermodel spread is of similar magnitude to the reanalysis spread. In particular, the

intermodel spread is over 40% of the climatological mean for the lower stratosphere and over 30% for the upper stratosphere. The annual cycle in the lower stratosphere has been linked to seasonality of wave forcing in the extratropics, subtropics and tropics (e.g. Randel et al., 2008; Ueyama et al., 2013; Ortland and Alexander, 2014; Kim et al., 2016). The semi-annual cycle in the upper stratosphere has been less studied. It is linked to the combined annual cycles of downwelling in each hemisphere (which have similar magnitude, in contrast with the lower stratosphere where the NH dominates). In addition, there is likely a

contribution from the secondary circulation associated with the Semi-Annual Oscillation (e.g. Garcia et al., 1997; Young et al., 2011), although this peaks at higher levels (∼0.1 hPa, Smith et al. (2017)).

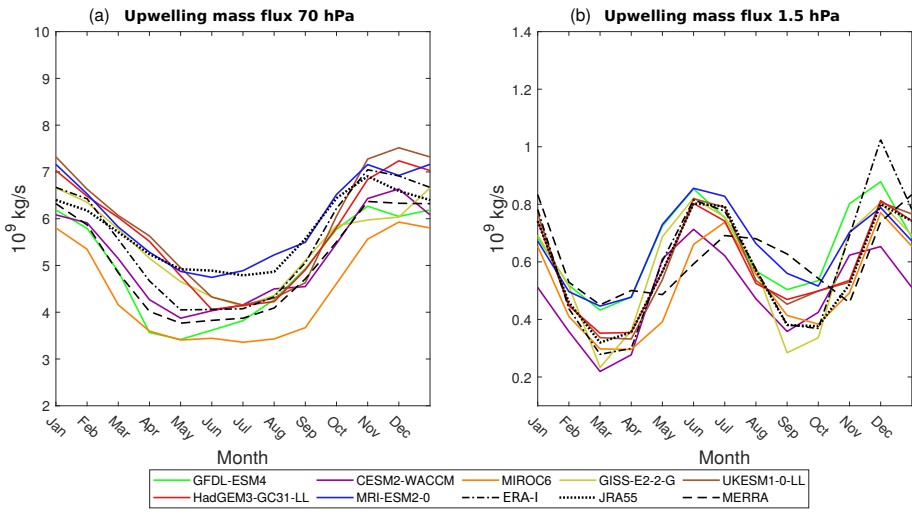

**Figure 2.** Seasonal cycle of tropical upwelling mass flux in the lower stratosphere (70 hPa, a) and in the upper stratosphere (1.5 hPa, b). Solid color lines show models and dashed black lines show reanalyses.

As mentioned in the Introduction, the mean age of air provides an estimate of the net transport circulation strength that can be compared to observational estimates. Figure 3 shows the AoA climatology at 50 hPa for the models that provide this quantity (see Table 1), together with the observational estimates described in Section 2. The simulated AoA values show considerable

spread across models, as previously shown for Chemistry-Climate Model Intercomparison project (CCMI) simulations (e.g., Dietmüller et al., 2017). The global mean age values vary by a factor of 2, between 2.5 and 5 years approximately. Nevertheless, the spread is within the large observational uncertainty. Note that the relationship between AoA and residual circulation strength is not straightforward. For example, the GFDL model features a weak upwelling, but the AoA is relatively young. In contrast, MRI has strong upwelling, but the AoA is the oldest. This lack of correspondence emphasizes the important role of mixing,

including subgrid effects, in determining the net transport strength (Garny et al., 2014; Dietmüller et al., 2017). The tropical leaky pipe model relates the net upwelling through an isentrope with the mass flux-weighted tropics/extratropics gradient in AoA (e.g. Linz et al., 2016). Linz et al. (2017) revealed a large discrepancy in the overturning circulation strength derived from the AoA gradient between the WACCM model and an older version of MIPAS observations, except at a level near 20 km. As a note, we applied a simple approximation of this method by computing the area-weighted age gradient on pressure levels (not

shown) and found a relationship with the net upward mass flux for the 1pctCO2 runs (3 models), but not for the historical runs (4 models). We therefore cannot extract robust conclusions due to the limited number of models providing AoA output.

### 3.2  Past trends

In this section we examine the BDC trends over the historical period, in particular over the last four decades. Figure 4 shows the multi-model mean trends in AoA (panel a) and vertical component of the residual circulation ($\bar{w}^*$, panel d) over the period

1975-2014. The AoA trends are negative everywhere with values around -0.1 years/decade, with trends in the lower strato-

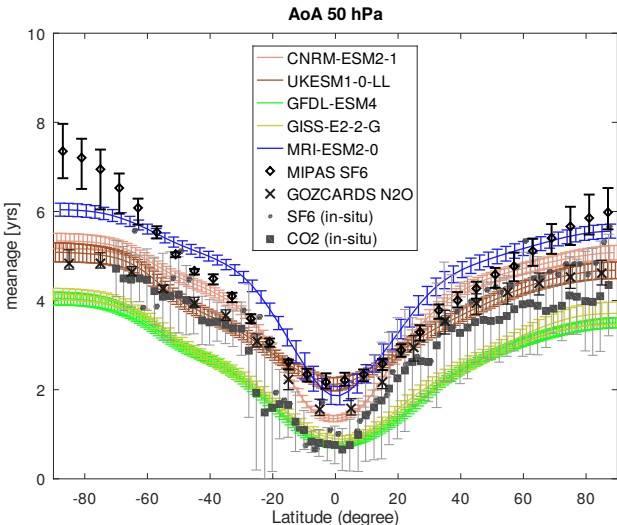

**Figure 3.** Annual mean AoA at 50 hPa for historical simulations averaged over 1990-2014, including the standard deviation of the annual means. Observationally derived AoA values are shown based on $SF_6$ measurements by MIPAS from 2005-2011 (Stiller et al., 2020), and $N_2O$ measurments from the GOZCARDS data set for 2004-2012 (Linz et al., 2017). For both, vertical bars represent the spread between minimum to maximum annual mean values. In addition, AoA values are shown as derived from in-situ measurements of $SF_6$ and $CO_2$ by Andrews et al. (2001). The errorbars for the in-situ measurements represent uncertainty in the measurement and derivation of AoA rather than interannual variability.

sphere larger in the Southern Hemisphere (SH) than in the Northern Hemisphere (NH). Consistently, the residual circulation accelerates throughout the stratosphere, with enhanced tropical upwelling and polar downwelling, strongest in the SH. Note that there is also reduced downwelling in midlatitudes in both hemispheres. The larger polar downwelling trends in the SH are consistent with recent results using CCMI models, and reflect the contribution of ozone depletion in the Antarctic lower

stratosphere to the BDC trends (Polvani et al., 2018, 2019; Abalos et al., 2019). In order to better capture this signal, the trends are shown separately for the end of the 20th century, a period of severe ozone depletion (panels b and e), and the beginning of the 21st century, when ozone depletion stops and its recovery starts (panels c and f). It is clear that the BDC trends are stronger during the ozone hole formation, particularly in the SH. The AoA trends for 1975-1990 are significantly different from those for 1998-2014 in the SH lower stratosphere and in the NH above 30 hPa and north of about 40°N (not shown). Note that

we have excluded the period 1991-1997 from the timeseries in order to avoid the influence of Pinatubo volcanic eruption on the trends. We caution that the trends are computed over short periods and the residual circulation presents high interannual variability, such that there is no statistical significance of the trends (panels d-f). Nevertheless, the influence of the ozone hole is clearly seen in the different trend magnitudes between the two periods, which are consistent across models.

While Figure 4 shows the general trend behavior for the multi-model mean, it is important to assess the robustness of

the trends across different members and models, especially given the relatively short periods under consideration. Figure 5

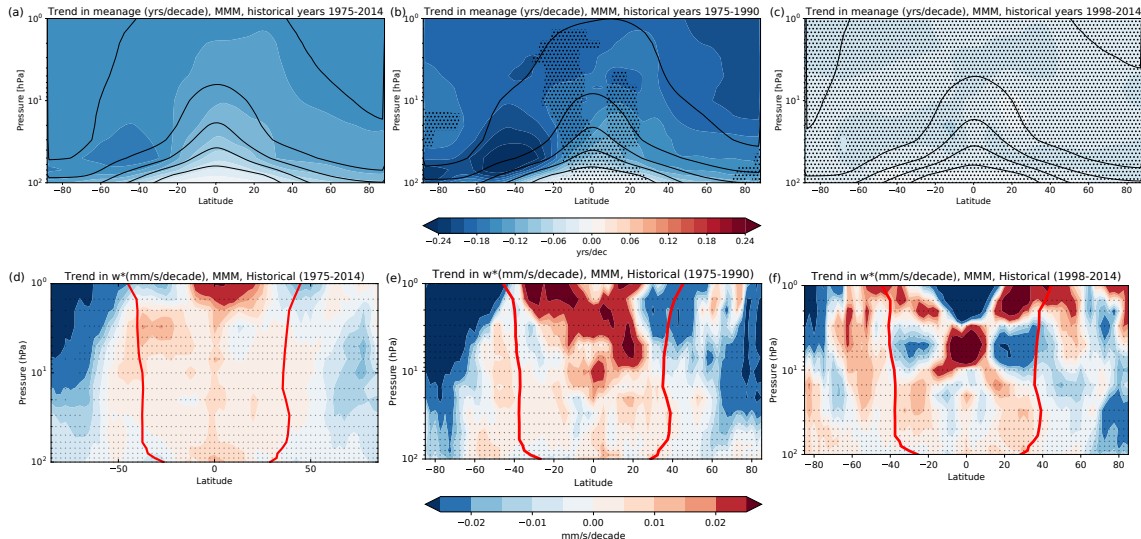

**Figure 4.** Multimodel mean linear trend in AoA (top panels, years/decade) and $\bar{w}^*$ (bottom panels, in $mm \cdot s^{-1} \cdot decade^{-1}$) from historical simulations in shading over years 1975-2014 (a and d) 1975-1990 (b and e) and 1998-2014 (c and f). In top panels the contours show the climatological AoA (years, contour interval 1 year, lower contour: 1 year). In the bottom panels the red contours show the turnaround latitudes averaged over each corresponding period. Stippling indicates statistically insignificant trends obtained with a Student's t test at 95% confidence level for more than 66% of the models.

examines the inter-model spread as well as the inter-member spread for each model, which quantifies the influence of internal climate variability on the trends. To do so, a histogram of trends for AoA and tropical upwelling is shown in Fig. 5 using all the members of each model. For upwelling, the same levels as in Fig. 2 are used in order to separate the shallow and deep branches. For AoA, the trends are shown over the regions with available trend estimates from observations, that is, the NH mid-latitude lower stratosphere (80 hPa) and mid-stratosphere (30 hPa). Observational trend estimates of AoA are shown for 80 hPa, and displayed in the legend for 30 hPa, as they are outside the range of the abscissa. In addition, the MMM trends from CCMI have been included in all panels. We do not include reanalysis trend estimates for upwelling because these show a larger spread than the models and thus do not help constrain the results (Abalos et al., 2015). In the lower stratosphere there is very good agreement between the CMIP6 MMM AoA trends and the observed trends (Fig. 5a). The trend in the CCMI MMM is also negative but not as strong. Note, however, the reduced number of models with AoA output in CMIP6. In the middle stratosphere (Fig. 5b) the MMM of both intercomparison projects produce a negative mean age trend between -2 and -3 %/$decade$, stronger for CMIP6. These values disagree strongly with the observed estimate of +3 %/$decade$, even when taking the large uncertainty into account. Even if one considers the updated estimate of AoA trends by Fritsch et al. (2020) of +1.5 $\pm$ 3 %/$decade$, the range of model estimates barely overlaps with the observational uncertainty range. The residual circulation trends in the shallow branch (Fig. 5c) range from 0.5 to 3.5 %/$decade$, with a maximum in the distribution slightly below 2 %/$decade$, consistent with previous climate model simulations. The CMIP6 MMM trend is in excellent agreement

with that from the CCMI MMM (Fig. 5c). In the middle stratosphere, the value of the CMIP6 MMM trend is similar to that in the shallow branch (slightly below $2\%/decade$), while the trend in the CCMI MMM is weaker (Fig. 5d).

When looking at individual simulations, the AoA trends show a similar spread in the trends across simulations of less than 3 $\%/decade$ at the two levels. In contrast, the residual circulation trends show a larger spread in the deep branch than in the shallow branch, with some members featuring slightly negative trends in the deep branch upwelling (trends range from almost -1 to over 5 $\%/decade$). Therefore, a deceleration of the deep branch over 1975-2014 is compatible with the internal variability in some of the CMIP6 models (although in the tail of the distribution), which could be consistent with observational AoA estimates. Nevertheless, we note that negative upwelling trends do not necessarily imply positive AoA trends, because the latter is an integrated quantity, affected non-locally by both advection and mixing (e.g. Garny et al., 2014; Linz et al., 2016). Indeed, the GISS and UKESM simulations that produce negative upwelling trends (panel d) still present a decrease in AoA in the middle stratosphere (panel b). Finally, note that the spread among individual models is comparable to the spread between ensemble members of one model both for AoA and upwelling and throughout the stratosphere. This highlights the vital role of internal variability for determining the trends.

We next examine the role of the limited sampling in the observational data on the detection of trends. Figure 6 shows timeseries and trends of AoA from the models subsampled at the locations and times of the Engel et al. (2009) measurements (though using monthly mean zonal mean output and averaging over the mean altitude of the measurements, 24-35 km). Also included are observational estimates of the AoA and its uncertainty from observations, both from Engel et al. (2009) and from the updated version from Fritsch et al. (2020), in which different parameters are used in the derivation of AoA from the tracer measurements. The latter study showed large sensitivity of AoA trends to assumed parameters in the derivation of AoA from non-linear increasing tracers. Here, AoA values derived with optimzed parameters are shown (using the convolution method and a ratio of moment of 1.25, for details see Fritsch et al. (2020). The subsampled trends are negative for every simulation, contrasting with the positive trends in the observational estimates (Fig. 6b). However, in this case the model trends are compatible with the observational trends from Fritsch et al. (2020) in 14 out of the 32 simulations (43%), for which the model and observational trend error bars overlap. It is important to point out that the uncertainties in the subsampled model AoA trends are 5 times larger on average than those using all model data, going from a 10% to a 50% uncertainty on average (not shown). These large errorbars due to subsampling are responsible for the agreement within uncertainties with observations obtained for some of the model simulations. Finally we note that, if only the $CO_2$ measurement locations are considered, the model trends are compatible with zero in most of the simulations (not shown). This is because the early observations before 1985, which are key to get negative trends in the models, are based on $SF_6$ (Fig. 6a).

To explore further the role of internal variability, we analyze how the trends depend on the length of the period considered. Hardiman et al. (2017) estimated the time of emergence of robust trends in tropical upwelling to be around 30 years. Here, we consider 18 ensemble members from the UKESM model, and compute the departure of the trends with respect to the ensemble mean, as a function of the length of the period (Fig. 7). The trends of AoA and upwelling are shown at the same regions as in Fig. 5. The results for AoA in the NH midlatitudes show that the trends agree within $\pm$ 30% with the ensemble mean for periods longer than $\sim$22 years (Figs. 7a and b). Given that the longest observational AoA estimates cover more than 30 years, natural

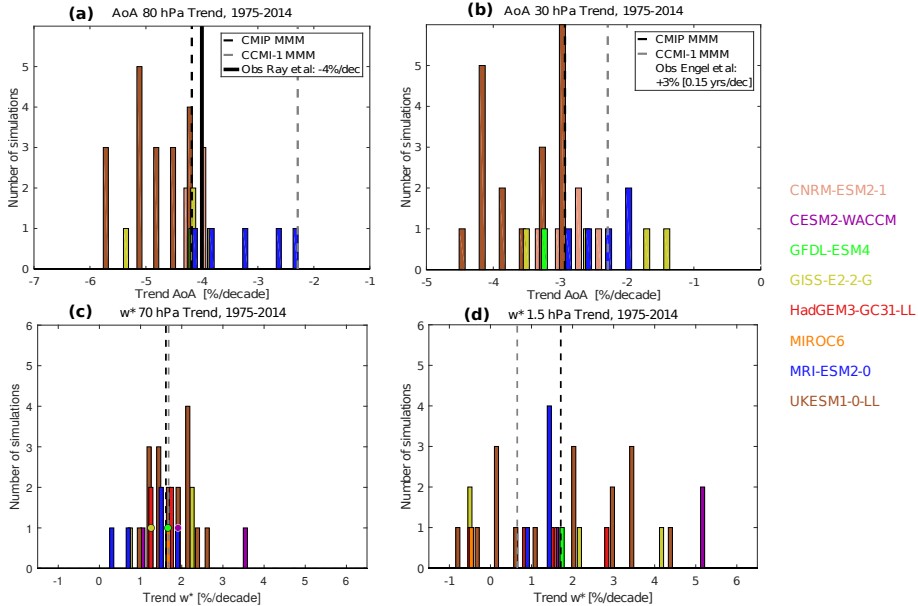

**Figure 5.** Absolute frequency distribution of trends (in $\%/decade$) over 1975-2014 from individual models and ensemble members in AoA at 80 hPa (a) and 30 hPa (b) averaged over 30-45N, and in tropical upwelling at 70 hPa (c) and 1.5 hPa (d). In panel (a) the observational trend estimate at 80 hPa over 1975-2012 from Ray et al. (2014) updated for the 2018 WMO Ozone Assessment report (Karpechko et al., 2018) is included as a black solid line. In panel (b), the observational trend estimate at 30 hPa over 1975-2017 from Engel et al. (2017) is only shown in the lagend, as the value is positive and thus outside the scale. Dashed lines show the MMM for CMIP6 models (black) and for CCMI models (gray). In panel (c) a colored circle is shown where a bar is covered by another model's bar. The number of available members from the historical simulations is: 5 for CRNM-ESM2-1, 3 for CESM2-WACCM, 1 for GFDL-ESM4, 4 for GISS-E2-2-G, 4 for HadGEM3-GC31-LL, 1 for MIROC6, 5 for MRI-ESM2-0-LL, 18 for UKESM1-0-LL.

variability cannot explain the discrepancy between the negative trends in the models and the positive trend in AoA derived from observations in the middle stratosphere. However, this argument assumes that the internal variability is realistically represented in the models, which is not necessarily true. When the same analysis is done for the residual circulation (Figs. 7c and d),

the trends need to be computed over longer periods to converge to the ensemble mean trend (more than 30 years), and even over 40 year periods the trends in different members converge only to within $\pm$ 50% of the ensemble mean, for the shallow branch (panel c), and to within $\pm$ 200% in the deep branch (panel d). At 10 hPa the 40-year trends show a $\pm$ 150% spread (not shown). These results highlight the substantially larger internal variability in the deep branch than the shallow branch of the residual circulation, and show that trends in AoA converge more rapidly to the MMM than those in upwelling. This is due to

the memory of AoA, being an integrated quantity in both time and space.

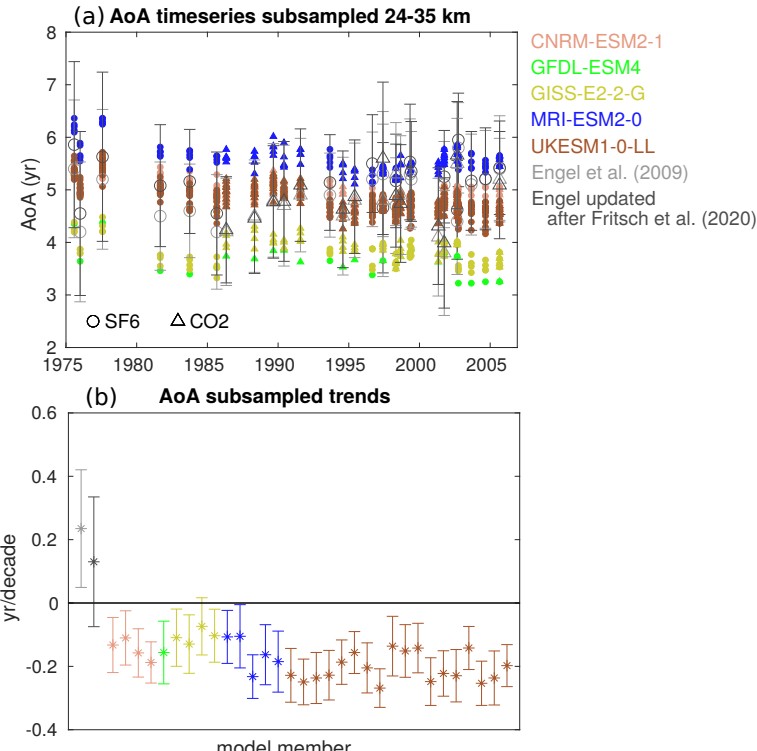

**Figure 6.** (a) Timeseries of AoA (in years) from individual model simulations sampled at the locations and times of the measurements from Engel et al. (2009), with different symbols representing the $SF_6$ and $CO_2$ measurements. Specifically, the zonal mean AoA has been averaged over the 24-35 km range and evaluated at the latitude and month corresponding to each measurement. Observational estimates from Engel et al. (2009) and updated after Fritsch et al. (2020) are included for comparison. (b) Trends in the timeseries in (a) for each model member and the observational estimates, with error bars corresponding to the 95% confidence level. The trends are computed for the combined timeseries considering both tracers. The individual measurement uncertainties are taken into account to calculate the trend error bar.

## 4   BDC response to $CO_2$ increase

In this section we examine the BDC trends and wave forcing in the 1pctCO2 simulations.

### 4.1   BDC trends

Figure 8 shows the trend in $\bar{w}^*$ for the 1pctCO2 simulations in the different models and for the MMM. This figure clearly

demonstrates the increasing strength of the residual circulation due to $CO_2$ increase, in both the deep and shallow branches. The trend is particularly strong in the lower stratosphere and near the stratopause, mirroring the climatological structure (Fig. 1). Changes in the turnaround latitudes indicate that the upwelling region narrows in the lower stratosphere in almost all models. These features are consistent with previous results (Palmeiro et al., 2014; Hardiman et al., 2014). In the upper stratosphere,

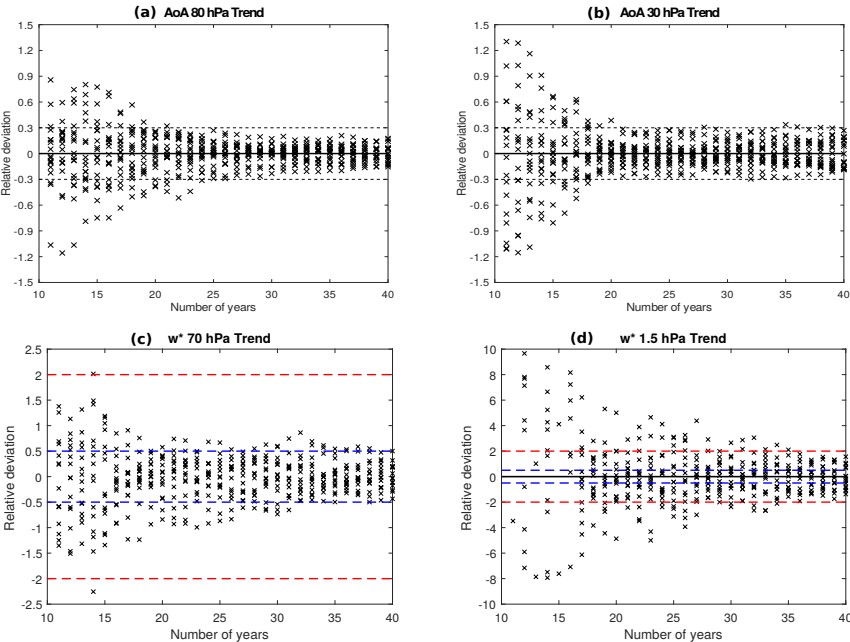

**Figure 7.** Trends from 18 ensemble members of UKESM1-0-LL for periods of increasing length (x-axis), ranging from 11 years (1975-1985) to 40 years (1975-2014). Trends of individual ensemble members (crosses) are displayed as relative deviations from the ensemble mean trend. (a) and (b) show trends in AoA and (c) and (d) show trends in upwelling, both variables at the same regions as in Fig. 5. Horizontal dashed lines mark fixed values to ease comparison across panels.

Hardiman et al. (2014) found a widening of the turnaround latitudes for CMIP5 MMM. They suggested that this change was
associated with a strengthening of the polar vortex in both hemispheres, which leads to reduced equatorward refraction of planetary waves. A more modest widening is found in the CMIP6 MMM, limited to the NH, perhaps linked to a strengthening of the polar vortex in the MMM for the subset of CMIP6 models used in the present study (not shown). Nevertheless, we note that the trends in the polar vortex are highly model dependent, and for instance the two models that show a clear widening of the tropical pipe in the NH upper stratosphere (CESM-WACCM and HadGEM) features opposite-sign trends in the polar
vortex. On the other hand, more detailed comparisons cannot be made since the forcings are different (RCP8.5 scenario in Hardiman et al. (2014) versus 1pctCO2 here). Despite the overall consistent structure of the trends, there is a notable spread in $\bar{w}^*$ trends, especially in the upper stratosphere (above 10 hPa), as will be quantified below. This is consistent with the large spread in the deep branch trends in the historical period discussed above. All models show stronger deep branch downwelling trends in the NH than the SH, and most models even feature slightly positive trends in the SH polar lower stratosphere. Such
asymmetry is not seen in the downwelling of the shallow branch over midlatitudes. A weakening of the polar downwelling in the SH was also seen in CCMI full-forcing simulations, linked to ozone hole recovery, which is however not present in the 1pctCO2 runs. Indeed, this weakening was not observed in CCMI sensitivity simulations in which ozone depleting substances did not change (Polvani et al., 2018, 2019). On the other hand, a weakened downwelling in response to increasing $CO_2$ is

consistent with an intensification of the SH polar vortex in response to greenhouse gas increase (McLandress et al., 2010;
Ceppi and Shepherd, 2019).

Similar to Fig. 8, Figure 9 shows the AoA trends in the different models and the MMM. The results show a consistent decrease in mean age throughout the stratosphere. In general there are weaker trends in the lower stratosphere in the SH than in the NH, consistent with weaker (and even opposite-sign) downwelling trends in this hemisphere seen in Fig. 8. There is substantial inter-model spread in the structure and magnitude of the trends. Common features include weaker trends in
the tropical pipe than at high latitudes, and particularly strong trends in the subtropical-midlatitude lower stratosphere. The stronger AoA trends in the extratropics as compared to the tropics and associated reduction in age gradient are consistent with the overturning acceleration, as revealed by the leaky pipe model (Neu and Plumb, 1999). This feature has also been linked to changes in mixing and to the upward shift of the circulation linked to tropopause rise (Oberländer-Hayn et al., 2016; Sacha et al., 2019).

Figure 10 explores the time dependence of AoA and upwelling trends in the 1pctCO2 runs. This is achieved by plotting trends for moving 30-year periods for each simulation to find out if trends are approximately constant or if they depend on the period under consideration. Note that, because there is no comparison with observations, we consider here the same regions for AoA and upwelling, representing the shallow and deep branches of the BDC. For individual simulations, the trends vary strongly for different periods. These variations are the largest for tropical upwelling in the deep branch, with several near-zero
and even negative trend periods (Fig. 10d). These large oscillations in the deep branch upwelling trends are reflected in the MMM. Note that the apparent periodicity of about 30 year is an artifact of the 30 year period used to compute the trends related with the Gibbs ringing (Gibbs, 1898), as we have checked by changing the length of the period (not shown). In contrast, the upwelling trend in the shallow branch is more consistently positive throughout the period, and shows an increase in the trend magnitude over time, from 2 to 4 %/decade in the MMM (Fig. 10c). The AoA trends are more similar at the two levels, with
consistent negative values throughout the period, despite the large oscillations (Figs. 10a and b).

Figures 5, 7 and 10 demonstrate the high sensitivity of trends in the deep branch residual circulation to the internal variability, as shown by the large inter-member spread and by the strong dependence on the length and starting year of the trend period. This contrasts with more stable and less uncertain trends in the shallow branch. They also reveal that AoA is a less noisy variable, featuring consistently negative trends in the deep branch across models and members, in contrast to the residual
circulation.

## 4.2 Drivers of the BDC trends

Figure 11 shows the contribution of the forcing from different waves to the vertical distribution of the net tropical upwelling for the MMM. To do so, the downward control principle has been applied between the annual-mean climatological turnaround latitudes (Palmeiro et al., 2014). As a novelty from previous multimodel assessments, here we examine the entire stratosphere,
including the forcing of the deep branch. The CMIP6 MMM results in Fig. 11a show that the climatological behavior of the tropical upwelling is driven fundamentally by resolved waves throughout the stratosphere. Their contribution is about 70% in the shallow branch, peaks in the middle stratosphere reaching 89% near 7 hPa, and is about 80% in the upper stratosphere.

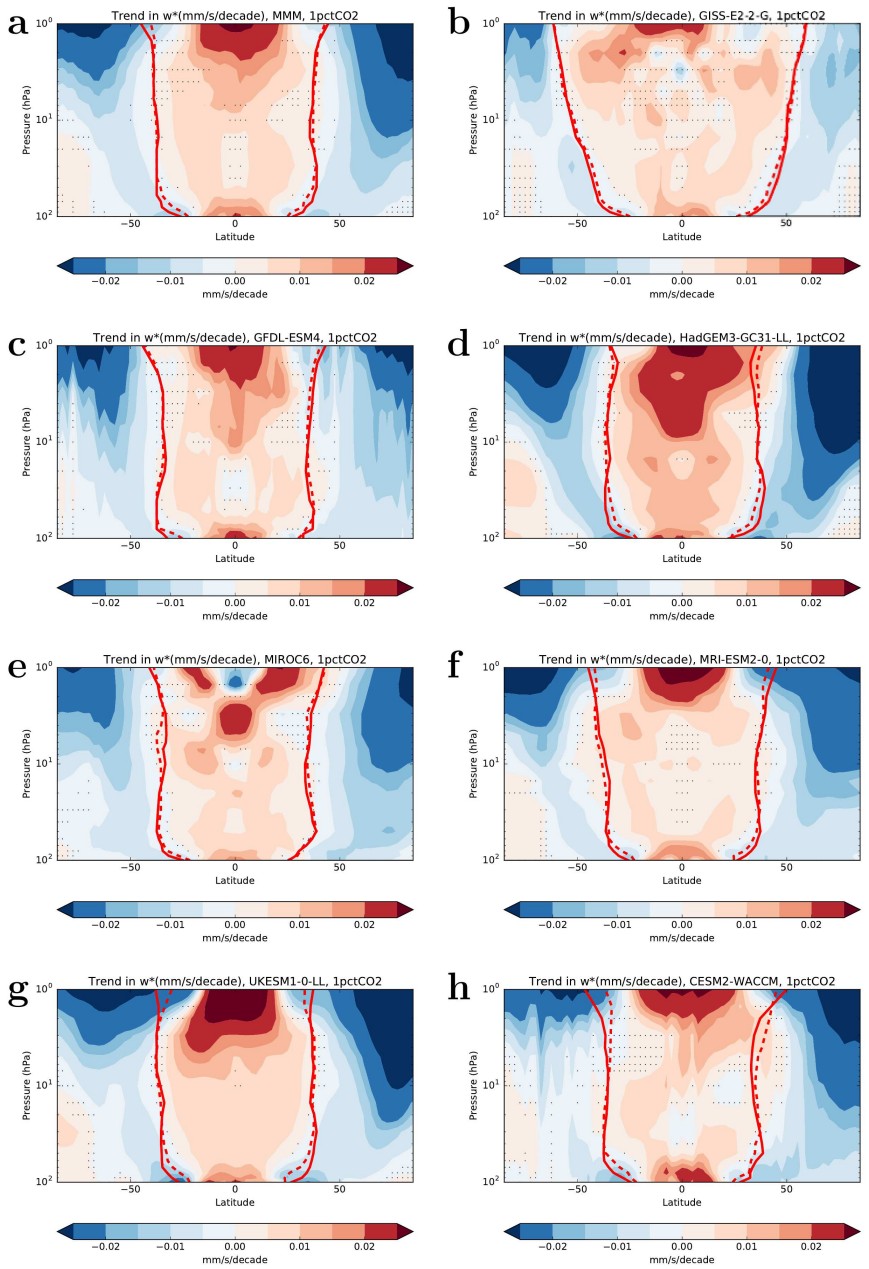

**Figure 8.** Trend in $\bar{w}^*$ ($mm \cdot s^{-1} \cdot decade^{-1}$), computed using 150 year time series from 1pctCO2 experiments. The top left panel shows multi-model mean (MMM). All other panels show individual models. Turn around latitudes show region of upwelling in first 20 years (solid red lines) and last 20 years (dashed red lines) of these 150 year simulations. Stippling in top left panel denotes regions where it is not the case that the trend is significant in at least 66% of models. Stippling in all other panels denotes regions where the trend in that model is not significant.

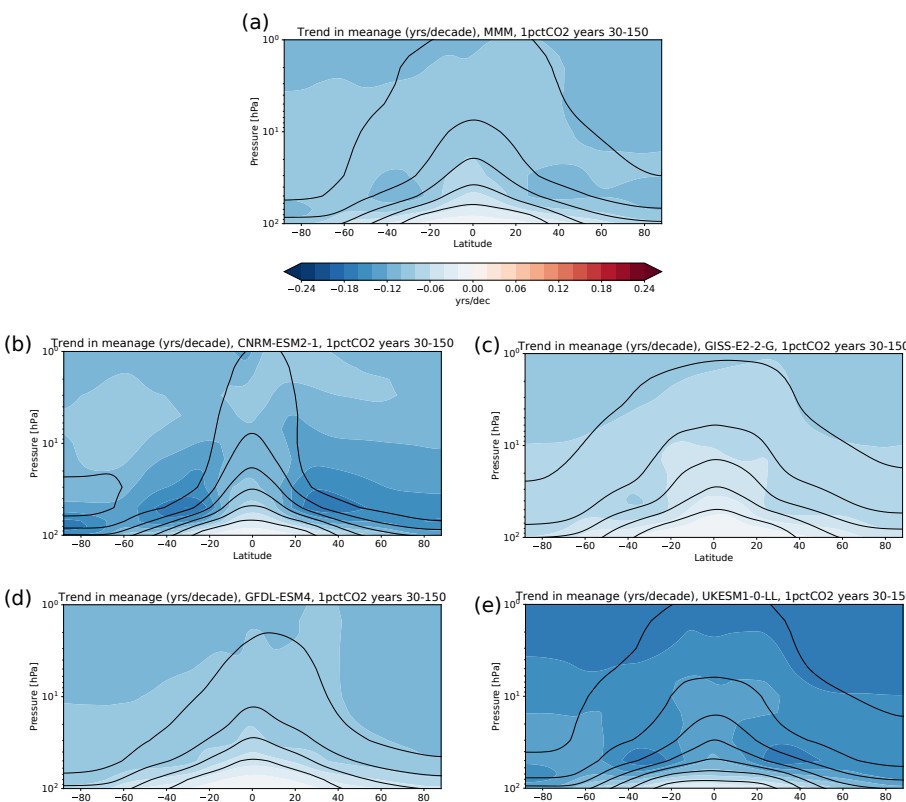

**Figure 9.** Multi-model mean linear trend in AoA from 1pctCO2 simulations over years 30-150 in years/decade (shading) and AoA climatological values averaged over the simulation years 30-60 (black contours, ci: 1 year, the lowest contour corresponds to 1 year). The first 29 years are not considered as the AoA needs time to spin up. Trends are statistically significant everywhere at the 95% level for all models.

Parameterized orographic gravity waves contribute 18% to the upwelling at 70hPa, in good agreement with previous multi-model studies (i.e. 20% in CCMVal2 models in Butchart et al. (2010)), and their contribution decreases at higher altitudes (6% at 1.5hPa). Parameterized non-orographic gravity waves (NOGW) are not negligible for the shallow branch and account for about 11% at 70hPa, while they become the second contributor in the upper stratosphere (15% at 1.5hPa). Note that model output for NOGW drag was available only for a small number of models in previous multi-model assessments, hampering a direct comparison.

As noted above, the vertical structure of the trends in upwelling (Fig. 11b) approximately mirrors that of the climatology (Fig. 11a). As shown in previous assessments (Butchart et al., 2010; WMO, 2014), resolved waves play the primary role in driving trends in the shallow branch. This is due to the intensification and upward displacement of the subtropical jets (not shown) and the upward displacement of the critical lines as discussed in other studies (i.e. Garcia and Randel, 2008; McLandress and Shepherd, 2009; Shepherd and McLandress, 2011; Hardiman et al., 2014). In particular, at 70hPa, the contribution to the total trend is 63% resolved waves, 25% OGWs and 11% NOGWs. In the upper stratosphere (above 10 hPa),

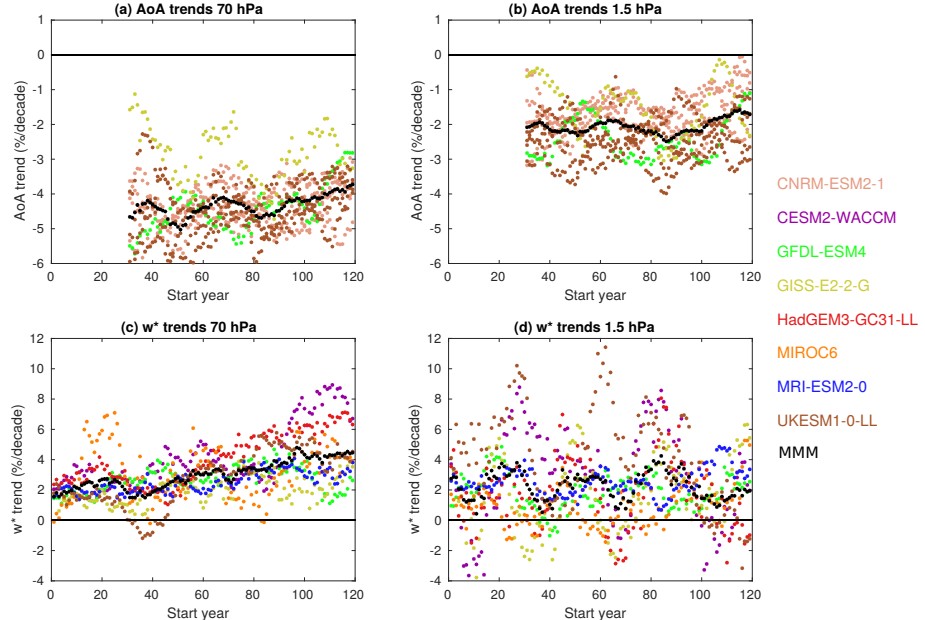

**Figure 10.** Trends in AoA (a and b) and upwelling (c and d) at 70 hPa (a and c) and 1.5 hPa (b and d) calculated from 30-years slices of the 1pctCO2 simulations with start year indicated at x-axis, for each individual simulation (dots, see legend for colors) and the MMM trend (black). AoA is averaged over 45°S-45°N and tropical upwelling is averaged between turnaround latitudes. Note that for upwelling only one member is included, but for AoA all available members are included, in order to have a more comparable total number of simulations for both variables. The number if available members is: 4 for CNRM-ESM2-1, 1 for GFDL-ESM4, 1 for GISS-E2-2-G, 4 for UKESM1-0-LL.

resolved waves and NOGW are equally important to the MMM trends, while the contribution from orographic gravity waves is much smaller. This is in agreement with the results of Palmeiro et al. (2014) for the previous version of WACCM, who explained the key role of NOGW due to changes in the filtering associated with changes in the background winds with increasing greenhouse gases. The resolved waves contribution peaks approximately at 7 hPa with a 59% for the MMM. At that level, NOGW contribute a 32% and OGW a 9%. At 1.5 hPa the percentages are 48%, 41% and 11%, respectively.

Figure 12 shows the intermodel spread in wave forcing. The forcing is more consistent across models for the climatology (panels a and b) than for the trends (panels c and d), in agreement with Butchart et al. (2010) results in the lower stratosphere (see also WMO (2014) and references therein). In the shallow branch, most models (except one) attribute the trends primarily to resolved wave drag. In addition, all the models show a relatively small contribution from NOGW (less than 20%). The contribution from OGW is more uncertain: it plays a significant role in 4 models (being the main forcing for GFDL-ESM4),

but is negligible in the other 3 models. Note that a present-day climatological source of NOGW is launched at 70 hPa in the extratropics in MIROC6 (Watanabe, 2008), which may explain the small contribution to the trends at this level and the negative contribution in the upper stratosphere.

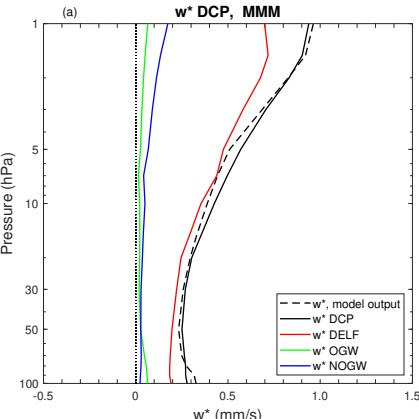
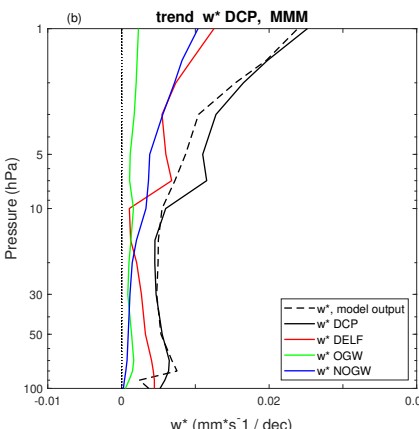

**Figure 11.** Climatology (a) and trends (b) of wave forcing of the residual circulation for the MMM, computed over the 150 years of the 1pctCO2 simulations. Dashed: Tropical upwelling provided as model output; black: total upwelling from downward control principle (DCP); red: upwelling due to resolved waves, computed from the Eliassen-Palm flux divergence (DELF); green: upwelling due to orographic gravity waves (OGW); blue: upwelling due to non-orographic gravity waves (NOGW). Note that the GISS model is not included because the NOGW output was not available.

The deep branch trends feature much larger spread, with a factor of 4 difference between the smallest trends (MIROC6) and the largest trends (UKESM) at 1.5 hPa. The larger role of NOGW compared to the shallow branch is clear, although there is quite a range in the contributions of resolved and NOGW. Four out of 7 models show comparable contributions, resolved wave forcing dominates in GISS-E2 and MIROC6 while in CESM2-WACCM there is a comparable contribution from NOGW and OGW, and a negligible contribution from resolved waves at 1.5 hPa. These results highlight a wide diversity of forcings of the $CO_2$-driven trends among models in the deep branch.

Note that there is no proportionality between climatological $\bar{w}^*$ and its trends across models, in contrast with results from Yoshida et al. (2018) for CMIP5 at 100 hPa. On the other hand, models with larger contributions from gravity waves in their climatology tend to have larger contributions in the trends.

## 5   BDC sensitivity to surface warming on different timescales

Figure 13a shows the sensitivity of the BDC to surface warming, calculated as the trends in upwelling mass flux (shallow and deep branches), relative to the trends in surface temperature (global and tropical). The results are shown for historical and 1pctCO2 simulations. While all models simulate a strengthening of the residual circulation in both shallow and deep branches along with surface warming in response to climate change, only the shallow branch is subject to a tight control of the global and tropical surface temperature. Note that the historical simulations trends are computed over the period 1960-2014, because this is when the surface warming is clearest. The trends over the period 1850-1959 (not shown) show a very large spread across models. The close connection between the residual circulation shallow branch and surface temperature is most evident

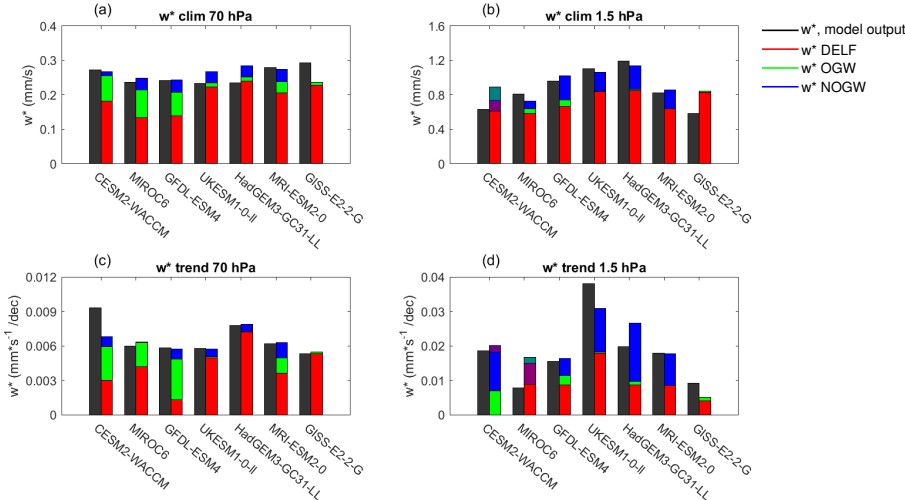

**Figure 12.** Decomposition into forcing from different waves of the tropical upwelling at 70 hPa (left) and 1.5 hPa (right) for the climatology (upper panels) and trends (lower panels) for the separate models, computed over the 150 years of the 1pctCO2 simulations. The black bar shows the upwelling directly from model output. The other colors are as in Fig. 11: red for resolved waves, green for OGW and blue for NOGW. The semitransparent shading in panels (b) and (d) indicates a negative contribution from a forcing, plotted from the top down in the corresponding color (e.g. NOGW in panel b for CESM2-WACCM). Note that for the GISS model NOGW output is not available, although these waves are present in the model.

in the last decades of the historical as well as in the 1pctCO2 simulations. The intermodel spread in the regression values reflects differences in the BDC sensitivity to surface warming among the models. For the shallow branch, the model sensitivity varies in the range from 5 to 13% per degree of surface warming. The strong connection between the shallow branch of the BDC and tropical surface temperature is consistent with the findings from previous works (Lin et al., 2015; Chrysanthou et al., 2020; Orbe et al., 2020). This statistical relationship reflects an underlying dynamical mechanism: tropical surface warming

leads to tropical upper tropospheric warming, which modifies meridional temperature gradients and thus wind shear (e.g. Garcia and Randel, 2008), altering the wave propagation and dissipation conditions (Shepherd and McLandress, 2011).

In contrast, the deep branch sensitivity to surface temperature is near zero in the historical runs, while in the 1pctCO2 runs it is small but consistently positive across the models (Fig. 13a). This difference between the two simulation types is likely due to the contribution of ozone depletion to the deep branch trends in the historical run, shown in Fig. 4. The impact of

ozone depletion on the residual circulation (present in the historical but not in the 1pctCO2 simulations) is independent from surface temperature, and therefore it reduces further the -already weak- connection between deep branch trends and surface warming. Note that the behavior is very similar when the global or the tropical mean surface temperature trends are considered. The disconnection between the deep branch of the residual circulation and surface temperature is consistent with the study of Chrysanthou et al. (2020), which finds that the acceleration of the deep branch is largely due to the direct $CO_2$ radiative effect.

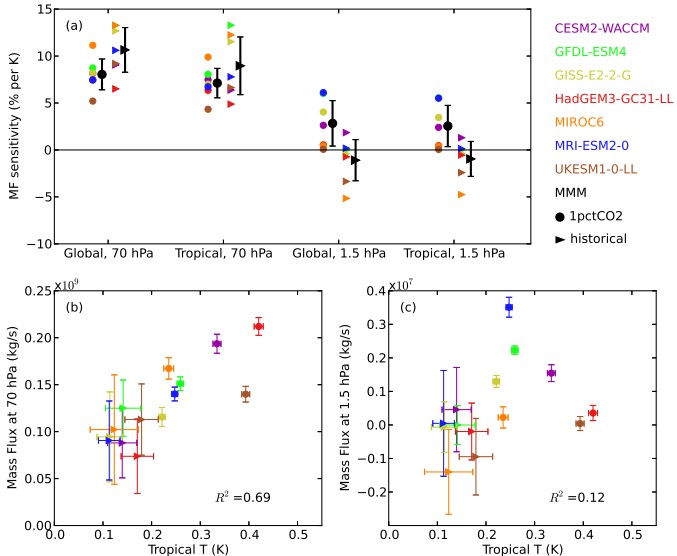

**Figure 13.** (a): Connections between long-term trends in mass flux in the shallow (70 hPa) and deep (1.5 hPa) branches and global and tropical mean surface temperature. The linear trends for the historical runs are calculated over the period 1960-2014 (triangles); for 1pctCO2 runs trends are calculated over years 1-150 (circles). Black errorbars show the MMM and one standard deviation among models. (b) and (c): Scatter plots of trends in mass flux at 70 hPa (b) and 1.5 hPa (c) versus tropical surface warming. The coefficient of determination $R^2$ across the models is shown in (b) and (c).

Panels b and c in Fig. 13 further demonstrate the connection between (tropical) surface temperature and lower branch trends (Fig. 13b), as well as the absence of such connection for the deep branch (Fig. 13c). Specifically, there is a strong correlation across the model simulations between the shallow branch and surface temperature trends ($R^2 = 0.69$), and a much reduced correlation for the deep branch ($R^2 = 0.12$). However, note that the high correlation disappears if only the historical simulations are considered. This is because the connection is mediated by the upper tropospheric warming, and there is substantial inter-model spread in the ratio of surface to upper tropospheric warming (see e.g. Po-Chedley and Fu, 2012). This spread only affects the historical simulations, as they have smaller warming trends than the 1pctCO2. The results in panels b and c are qualitatively similar for the global surface temperature trends (not shown).

The connection between surface temperature and BDC residual circulation is explored at interannual and decadal timescales in Figure 14. The decadal regression coefficients (panel b) show a very high consistency with the trend behavior in Fig. 13, both qualitative and quantitative. The corresponding coefficients of determination ($R^2$, panel d) reveal that the fraction of variance of the shallow branch tropical upwelling explained by the surface temperature is around 35-60%, with a wide inter-model spread (ranging from 15 to 85%). These values are about 15% higher for the 1pctCO2 than for the historical simulations, and also for the tropical than for the global surface temperature.

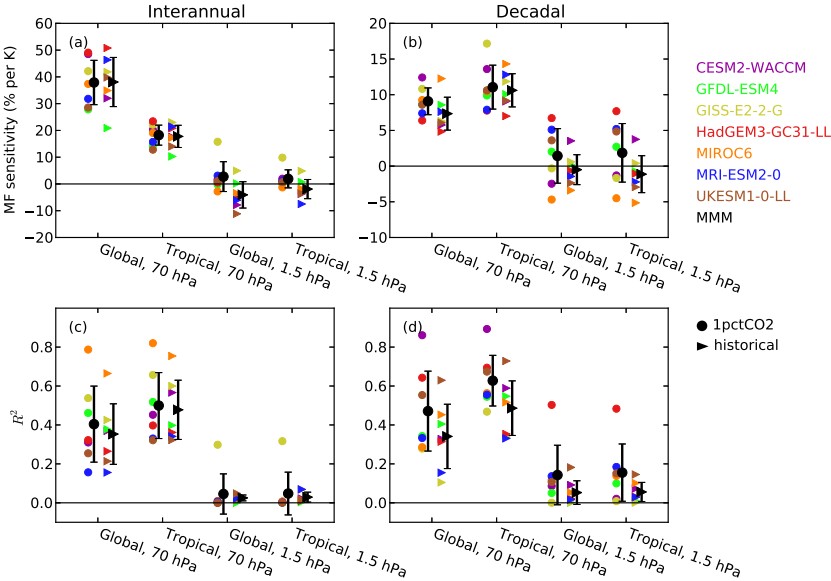

**Figure 14.** Regression (top) and determination (bottom) coefficients between mass flux in the upper and lower stratosphere with global and tropical mean surface temperature on interannual (left) and decadal (right) timescales. Interannual variability is obtained as the annual mean minus 5-year running mean data, decadal variability is the detrended 5-year mean. Black errorbars show the MMM and one standard deviation among models. Triangles denote output from historical runs and circles from 1pctCO2 runs.

The explained variances ($R^2$) for interannual variations show similar values to the decadal timescales, and are comparable for the tropical as for the global surface temperature. This implies that the surface warming signal on the lower branch of the residual circulation is controlled primarily by tropical temperature. However, the regression values for the shallow branch are notably higher (approximately twice as large) for global than for tropical surface temperatures (Fig. 14a) in the case of interannual variability. This likely reflects the fact that interannual temperature variations often have an antisymmetric pattern between tropics and extratropics. In particular, the connection between the residual circulation and surface temperature on interannual timescales is dominated by ENSO (e.g., Calvo et al., 2010), which features strong sea surface temperature anomalies (SST) confined to the tropics, and opposite sign anomalies in the extratropics. The globally averaged surface temperatures attenuate the signal (associated with ENSO) that is mainly responsible for interannual variability in the mass flux. This results in higher regression coefficients for the global temperature. In contrast, long-term trends and low-frequency variability have a more uniform latitudinal pattern.

## 6   Conclusions and outlook

The present paper examines the BDC in CMIP6 models, focusing on the residual circulation and the mean age of air.

First, historical simulations are used to compare the climatology and past trends to observations and reanalyses. The climatological average and seasonality of the BDC in CMIP6 models lie within the spread of observational (and reanalysis) estimates. However, there is a large spread in the magnitude among models, which is as large as that across observational estimates. The BDC trends in historical simulations are stronger during the ozone depletion period than after, which reflects the important contribution of ozone depletion to BDC acceleration shown in previous chemistry-climate model studies. There is very good agreement between models and observations in the shallow branch trends. In contrast, consistent with previous studies, there remains a clear disagreement in the AoA trends between models and observations in the middle and upper stratosphere, with models featuring robust negative trends. Even when the model's AoA is subsampled in space and time as the observations, the model trends remain negative in the middle stratosphere for all available models and ensemble members (Fig. 6). Nevertheless, in this case a substantial fraction (over 40%) of the available simulations the model and observed error bars overlap with the observational estimate updated in Fritsch et al. (2020). It is important to note that, in addition to the sampling, another caveat in the comparison between observational and model AoA data are differences in the properties of idealized versus realistic tracers, as discussed in detail by Garcia et al. (2011). The trends in the deep branch of the residual circulation reveal a large spread among models and across different members of the same model (Figs. 5, 7). In particular, the inter-member spread in deep branch residual circulation trends is about $\pm 200\%$ of the ensemble mean trends, even when considering long (40 year) periods. In contrast, for the shallow branch the spread is 4 times smaller. This reveals a notably stronger influence of internal variability on the deep branch than the shallow branch trends. Importantly, the trends are more robust for AoA than for upwelling, with inter-member spread in the trends below $\pm 30\%$ for periods slightly longer than 20 years, both in the lower and middle stratosphere.

The sensitivity of the BDC to $CO_2$ increase is examined, and the robustness of the trends and their wave forcing are explored. In contrast with previous BDC analyses based on multi-model assessments with CCMVal, CMIP5 and CCMI, we focus here on the response to $CO_2$ alone, using the 1pctCO2 simulations. All models produce stronger residual circulation acceleration in the NH than in the SH, and it actually decelerates in the SH polar lower stratosphere, possibly due to the $CO_2$ effects on the polar vortex discussed in recent works (e.g. Ceppi and Shepherd, 2019). An analysis of the wave forcing of the residual circulation shows that shallow branch forcing of climatology and trends is mainly due to resolved waves with a contribution from OGW, consistent with previous studies. For the deep branch, the main drivers of climatology and trends are resolved waves and parameterized NOGW, but there is a wide spread across models, especially for the trends. There is a very large inter-model spread in deep branch trends (a factor of 4), which could be linked to the spread in parameterized gravity wave forcing. In contrast, the spread in the shallow branch trends is less than 30%. The uncertainty in deep branch residual circulation trends is emphasized in the large multi-decadal fluctuations found over the 150 simulation years. On the other hand, the shallow branch trends are found to increase over time with $CO_2$ increase, by approximately a factor of 2 for the MMM. In contrast, the AoA trends are more robust over time, consistent with the results for the historical simulations.

Finally, the connection between surface temperature and the BDC is investigated. We find a strong connection between the shallow branch of the residual circulation and the tropical and global surface temperature. Long-term trends in lower stratospheric upwelling feature a sensitivity of 7-10% per degree of surface warming in the models (Fig. 13). On interannual

and decadal timescales, surface temperature explains 35-60% of the shallow branch variance on average (Fig. 14). Note that the strong connection of shallow branch acceleration with surface warming is consistent with the correlation with the upward shift of the tropopause pointed out by (Oberländer-Hayn et al., 2016). In contrast, the deep branch variability is not correlated with surface temperature on any timescale.

One of the key results of the present paper is the difference between shallow and deep branches of the residual circulation. The CMIP6 models confirm that, while trends in the shallow branch can now be reconciled with observations (Fu et al., 2015; WMO, 2018), much larger uncertainties remain for the trends in the deep branch. We note that, while a robust mechanism for the acceleration for the shallow branch has been described (Shepherd and McLandress, 2011), the drivers of deep branch acceleration remain largely unexplored. Previous studies point to the effects of stratospheric zonal wind trends on the filtering of NOGW. The CMIP6 model results in the present paper confirm the important role of NOGW for the deep branch trends. The zonal mean wind trends show acceleration of the polar jet in both hemispheres for the MMM of the CMIP6 model subset used in this study, but there is a large spread in the trends in the NH, with some models featuring deceleration (not shown). This is consistent with large differences in resolved versus NOGW forcing contributions to the deep branch among models. However, the compensation mechanism (Cohen et al., 2013; Sigmond and Shepherd, 2014) implies that the different relative contributions from resolved versus parameterized gravity waves does not necessarily lead to differences in the net residual circulation trends. Overall, open questions remain regarding the deep branch trends and their forcing mechanisms. Reducing uncertainties in deep branch trends is particularly relevant to better constrain the future distribution of ozone in the polar stratosphere, affected not only by direct transport but also by the descent of ozone-depleting chemical compounds from the mesosphere (Maliniemi et al., 2020).

The results show that AoA is a much less noisy variable than $\bar{w}^*$, implying that robust trends could be extracted from relatively short periods (20 years). The advective circulation can be approximated from AoA using the leaky pipe model, in order to compare with observations, as done in Linz et al. (2017). Unfortunately, the global AoA observational estimates available are not long enough to evaluate trends. In addition, it is crucial to account for the large uncertainties in deriving AoA trends from realistic tracers (Fritsch et al., 2020). We note that, for the few models providing both AoA and $\bar{w}^*$ (3 models for 1pctCO2 and 4 for historical simulations), it is not possible to extract robust conclusions on the relationship between the strength of the climatological values or the trends of the two variables. Establishing relations between these two magnitudes would help evaluate the spread in the magnitude and variability of mixing across models (e.g., Dietmüller et al., 2017; Eichinger et al., 2018). Based on the results of the present study that highlight the robustness of AoA trends, we suggest that AoA should be a first-priority consistently defined diagnostic for the next CMIP project (Gerber and Manzini, 2016).

*Data availability.* CMIP6 output is available online at various sites listed at https://pcmdi.llnl.gov/CMIP6/.

*Author contributions.* MA wrote the manuscript with input from all coauthors. MA, NC, SB-B, HG, SCH and PL designed the article and/or produced figures. MBA, NB, RG, CO, DS-M, SW and KY contributed to the manuscript and provided model expertise.

*Competing interests.* No competing interests are present.

*Acknowledgements.* We are thankful to Gabi Stiller for providing the latest version of MIPAS data, as well as to Ed Gerber and two anonymous reviewers for insightful and constructive comments on the paper. MA acknowledges funding from grant CGL2017-83198-R (STEADY) and from the Program Atracción de Talento de la Comunidad de Madrid (2016-T2/AMB-1405). NC was supported by the Spanish Ministry of Science, Innovation and Universities through the JeDiS (Jet Dynamics and extremeS) project (RTI2018-096402-B-I00). SB-B acknowledges the FPU program from the Ministry of Universities, grant number FPU19/01481. HG was funded by the Helmholtz Association under

grant VH-NG-1014 (Helmholtz-Hochschul-Nachwuchs-forschergruppe MACClim). PL acknowledges award NA18OAR4320123 from the National Oceanic and Atmospheric Administration, U.S. Department of Commerce. SW was supported by the Integrated Research Program for Advancing Climate Models (TOUGOU) Grant Number JPMXD0717935457 and JPMXD0717935715 from the Ministry of Education, Culture, Sports, Science and Technology (MEXT), Japan. The Earth Simulator at the Japan Agency for Marine-Earth Science and Technology (JAMSTEC) was used for the MIROC6 simulations. The work of SCH, MBA and NB was supported by the Met Office Hadley Centre

Climate Programme funded by BEIS and Defra. CO acknowledges the NASA Modeling, Analysis and Prediction program and resources provided by the NASA High-End Computing (HEC) Program through the NASA Center for Climate Simulation (NCCS) at Goddard Space Flight Center.

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
