# Peer review of "The Brewer-Dobson circulation in CMIP6"

_Atmospheric Chemistry and Physics, 2021_

## Community Comment (CC2)

Comment on **"The Brewer-Dobson circulation in CMIP6**" by Abalos, M. et al., ACPD, 2021

Dear authors, thank you very much for this interesting study on the Brewer-Dobson circulation in the new CMIP6 simulations. Going through the manuscript we noted 2 points, where our research can help to interpret your results. Please see below our two points.

Best wishes
Petr Sacha and Roland Eichinger.

 **- POINT 1:**
 L212: "*Common features include [...]* **particularly strong** *trends in the subtropical-midlatitude lower stratosphere.*"

Please note that the existence of these regions has been pointed out and studied in detail in Šácha et al. (2019, ACP). These trend patterns can serve as a visual proxy for structural changes in the lower stratosphere in the models.

**- POINT 2:**
Please clarify your methodology with respect to the usage of w*_bar (results around Figs. 7, 10 and 11).
As reported in the supplement of Dietmüller et al. (2018) for CCMI simulations, there were inconsistencies in the type of w*_bar provided by the modelling groups, despite the log-pressure formula being solicited in the data request.
In the DynVar data request by Gerber and Manzini (2016), the log-pressure formula is also solicited. If there are inconsistencies in the w*_bar formulae between CMIP6 simulations, this can result in differences in wstar climatology and trends as quantified in Eichinger and Sacha (2020). Hence, our findings can help to narrow down the w*_bar differences in Fig. 11. Generally, note that due to stratospheric cooling, the relation of log-pressure metres to geometric metres is not constant, which projects also to the magnitude of w*_bar trends.

REFERENCES:

- Šácha, P., Eichinger, R., Garny, H., Pišoft, P., Dietmüller, S., de la Torre, L., Plummer, D. A., Jöckel, P., Morgenstern, O., Zeng, G., Butchart, N., and Añel, J. A.: Extratropical age of air trends and causative factors in climate projection simulations, Atmos. Chem. Phys., 19, 7627–7647, https://doi.org/10.5194/acp-19-7627-2019, 2019.
- Eichinger, R, Šácha, P. Overestimated acceleration of the advective Brewer–Dobson circulation due to stratospheric cooling. *QJR Meteorol Soc.* 2020; 146: 3850– 3864. https://doi.org/10.1002/qj.3876
- Dietmüller, S., Eichinger, R., Garny, H., Birner, T., Boenisch, H., Pitari, G., Mancini, E., Visioni, D., Stenke, A., Revell, L., Rozanov, E., Plummer, D. A., Scinocca, J., Jöckel, P., Oman, L., Deushi, M., Kiyotaka, S., Kinnison, D. E., Garcia, R., Morgenstern, O., Zeng, G., Stone, K. A., and Schofield, R.: Quantifying the effect of mixing on the mean age of air in CCMVal-2 and CCMI-1 models, Atmos. Chem. Phys., 18, 6699–6720, https://doi.org/10.5194/acp-18-6699-2018, 2018.

---

## Author Response (AR1)

We are thankful to Ed Gerber and the two anonymous reviewers for the thorough revision and insightful comments, we believe that by addressing them we have improved the paper notably.

**REVIEWER 1**

The study by Abalos et al analyzes the Brewer-Dobson (BD) circulation in a subset of CMIP6 models for which relevant diagnostics are available. Both ability of the models to reproduce the observed features of the BD circulation as well as their future projections are analyzed. The main novelty of the study is that the results found in the previous generations of CMIP models, and other chemistry-climate model evaluations (e.g. CCMVal) such as acceleration of the BD circulation in response to global warming, are confirmed with the new generation of the models. On the other hand, the study also highlights considerable uncertainty in the quantitative estimates of the BD trend, both historical and predicted future trends. The uncertainty arises from the internal variability, but also from the intermodel differences. The latter is evident, in particular, from the fact that the models disagree about the driving forces of the trends.

I recommend the paper to publications and ask the authors to clarify some points listed below:

1. The use of CMIP6 data:
The authors use 1 member per model however I don't see a justification for this choice. There is considerable internal variability in BD diagnostics and, in order to get a better quantification of the signal, in particular the BD trends, I think all members should be considered.

For the climatological figures it is irrelevant to use one or more members, since the climatology coincides in all. For the trend figures, we deliberately decided to use one member because there is a very uneven number of members for the different models (from 1 to 18), so if we would show the ensemble mean trends the figure would be severly biased towards the behavior of the models with a largest number of trends.
However, we do use all available members for the trends in Figs. 5, 6, 7 and 10 (for AoA). In these figures we explicitly examine the role of or internal variability on the BDC trends by examining intermember spread. Therefore, we do not use ensemble mean to avoid biasing the results, but we do exploit the multiple members when available to examine internal variability.

We have modified the manuscript including this text:
" We use one member of each simulation because there is a very uneven number of members for the different models (from 1 to 18), and therefore the comparison across models would be unfair if the ensemble mean were used for each model. Nevertheless, we do exploit the multiple members when available in order to explore the role of internal variability on the trends (Figures 5, 6, 7 and 10). "

We have also modified the figure captions of Figs. 5 and 10 to include the information on the available members for each model.

2. Significance of w* historical trends:
This analysis is problematic. The authors conclude that w* historical trends shown in Fig. 4 are insignificant; however, looking at Fig.5c where all individual simulations show negative trends, I wonder whether the trends are indeed insignificant. I would expect that by averaging across many members all of which agree on the sign of the trend, the internal variability would be reduced, and the signal would emerge. Further, if w* trends are

indeed insignificant then how one can see an influence of ozone hole (L137)? This need to be clarified.

The trends in Fig. 4 are insignificant due to the large internal variability in this variable (w*). Figure R1 shows the trends averaged for the ensemble mean in the models that provide more than one multiple members: a) HadGEM3: 5 members, b) UKESM1: 18 members, c) CESM2-WACCM: 3 members. Still, the trends remain mainly insignificant at the 95% confidence level, except in the SH high latitudes upper stratosphere and in a small region in the tropical lower stratosphere. Panel (d) shows the trends in the MMM, using only 1 member as in Fig. 4, to avoid biasing the signal towards the models with more ensemble members, as argued above. The figure shows some significance only in the SH upper stratosphere, and we have now pointed this out in the manuscript (Fig. 4 caption). However, as the reviewer points out, the confidence in the trends comes from the agreement in sign across the models. This applies to the difference in trends between the ozone deletion and recovery periods: although the trends are statistically insignificant, there is a clear change in the magnitude of the trends consistently seen in all models. In addition, the difference in the trends between the two periods is statistically significant in the tropical and SH upper stratosphere (not shown). This is already pointed out explicitly in the manuscript.

[Figure]

Figure 1R. Panels a-c: Trends in ensemble mean residual circulation w* for the models with more than one member. Panel d): Trends in MMM residual circulation w* considering one member. The dots indicate lack of statistical significance at the 95% level.

Other comments:

L14: Perhaps you can refer to specific Chapter of the WMO report rather than to the whole

report, here and in the other places? The reports are large, and the reader does not necessarily know which chapter to look at.

Done, Chapter 5 is cited now.

L34: I think it is enough to write either "non-significant positive trends" or "slightly positive trends" depending on what you want to emphasize.

Changed to "non-significant positive trends".

L45: "response to an IDEALIZED 1%/year CO2 increase"

Added.

L105: Fig.2 shows that MRI and JRA55 are nearly identical, and both represent the higher bound of the model spread in the lower stratosphere. Given that both, MRI and JRA55, are based on the JMA operational model (I believe so), would that indicate that the influence of assimilation on BDC is negligible, at least in JRA-55? Would this explain the spread across the reanalyses?

Certainly, both MRI-ESM2.0 and JRA-55 are derived from JMA operational model, although many physical parameterization schemes (e.g., cumulus convection, cloud microphysics, and nonorographic gravity wave drag) and processes (e.g., interactive/prescribed chemistry and aerosol) are different. JRA55 simulations without assimilation are provided as JRA55AMIP. Kobayashi & Iwasaki (2015) assessed the Brewer-Dobson circulation in JRA55 and JRA55AMIP. Fig. 6 in Kobayashi & Iwasaki (2015) shows that 70 hPa upward mass flux in JRA55AMIP is roughly 30% smaller than JRA55. Thus, upward mass flux at 70 hPa in JRA55 is not equal to that in MRI-ESM2.0, unless any data assimilation is performed.

Figure 4: Do contours for AoA climatology in panels a-c start from 1 year?

Yes, we have added this information to the caption.

L162: The models simulate an acceleration of the BD circulation over 1975-2014, not deceleration. Or?

Yes, in general most simulations show an acceleration of the residual circulation. However here it is emphasize that a few simulations show negative trends.

L194: "strengthening of the polar vortex ... leads to reduced equatorward refraction of planetary waves" Fig. 3 from Hartmann et al (2000) shows that a strengthened polar vortex leads to an enhanced equatorward refraction of planetary waves, contrary to what the authors state. Also, given the spread in the widening across individual models, you could analyze the relationship between the changes in the turnaround latitudes and changes in the polar vortex across the models. At least for WACCM the mechanism does not seem to work, because this model simulates both poleward shift of the turnaround latitude (Fig. 7h) and vortex weakening (my own calculations). So, I am not convinced the proposed mechanism is valid.

After this reviewer's and Rev. 3 comment we have checked that there is not much consistency across models in the correspondence between turnaround latitudes and polar vortex trends. We have accordingly change the text to reflect this (L220): "A more modest widening is found in the CMIP6

MMM, limited to the NH, perhaps linked to a strengthening of the polar vortex in the MMM for the subset of CMIP6 models used in the present study (not shown). Nevertheless, we note that the trends in the polar vortex are highly model dependent, and for instance the two models that show a clear widening of the tropical pipe in the NH upper stratosphere (CESM-WACCM and HadGEM) features opposite-sign trends in the polar vortex."

Figure 9: Why AoA panels start from year 30? I understand that x-axis shows starting year, so why not start from year 0 as is done in w*?

AoA is computed from a passive tracer in the models, which is initialized in the troposphere and needs to be transported to fill out the atmosphere before AoA can be computed. This is why we do not present trends computed over the first years of the simulations for this variable.

Figure 9 caption: "... a comparable total number of simulations for both VARIABLES" (not magnitudes)

Changed.

Figure 11: I cannot understand how you indicate negative contribution with semitransparent shading, I am sorry. Is there any other way to draw it?

We changed the figure by plotting the semitransparent shading in the color corresponding to the negative forcing. This representation provides a clearer view of the contribution from the different forcings as well as providing a direct comparison between the w* provided by the model and the upwelling resulting from downward control calculation. In addition, to make the figure clearer we have added an example in the Figure caption.

Reference: Hartmann, D. L., J. M. Wallace, V. Limpasuvan, D. W. J. Thompson, and J. R. Holton (2000), Can ozone depletion and global warming interact to produce rapid climate change? Proc. Natl. Acad. Sci. U. S. A., 97, 1412– 1417, doi:10.1073/pnas.97.4.1412

**REVIEWER 2 (Ed Gerber)**

The authors document the climatology of Brewer-Dobson Circulation in CMIP6 models, and its response to forcing in the historical and 1pctCO2 doubling integrations. They contrast the behavior of models with available observations/reanalyses, and provide a process oriented exploration of the residual circulation, breaking down the role of resolved waves vs. parameterized gravity waves. This is the first time CMIP class models have provided the necessary output for this analysis, and I expect this paper to become an important reference point for our understanding and discussion of the BDC. I therefore strongly recommend publication of this thorough and well written manuscript pending consideration of the minor suggestions below.

I hope that authors see my suggestions below as a genuine attempt to help improve the paper. This is a very strong manuscript, and I very much support its publication.

Ed Gerber

General minor suggestions

1) I feel there was tension, starting from the abstract, about the narrative on the comparison of observations and models, particularly in the upper stratosphere. For instance, at line 4 of the abstract suggest that the models are inconsistent, but then immediately following, at line 6, it is suggested that there is great uncertainty in the model trends. I am not an expert in the observed trends, and my main suggestion is chiefly to be more consistent with the message. Do the authors mean something like "while there is great uncertainty in trends in the upper branch of the BDC in models, model trends appear to be statistically distinguishable from observed trends"? If this is the case, I would first highlight the uncertainty, and then state that despite this great uncertainty, models cannot be reconciled with observations.

These two sentences refer to different simulations (historical and 1pctCO2); only the historical runs can be compared to observations so the inconsistency refers to them, while the second refers to increase CO2 response. The trends in the historical runs are all negative (including error bars), while the observations suggest positive or zero trends. However, as now shown in the new Fig. 6 (see below), there is an overlap when considering the large observational uncertainty, in particular the results of Fritsch et al 2020. In order to include information on the role of observational uncertainty in the trends, we have added the following sentence to the abstract after the inconsistency sentence: "Nevertheless, the large uncertainty in the observational trend estimates opens the door to compatibility. In particular, when the limited sampling of the observations is considered, model and observational trend errorbars overlap in 40\% of the simulations."

And this said, I continue to worry that the uncertainty in observed trends may be underestimated. Am I correct that the key mismatch is with Engel et al. 2017 (Air Core measurements at two sites) and MIPAS retrievals from Stiller et al. (2012, 2020), though the MIPAS estimate has become closer with the revision of our treatment of SF6 (Fritsch et al. 2020).

As highlighted, for example, by Garcia and Randall (2011), there are significant uncertainties associated with the fact that observation estimates (as with air core samples) are based on sparse measurements relative to the model based estimates using global averages.

https://journals.ametsoc.org/view/journals/atsc/68/1/2010jas3527.1.xml

To be constructive, I am curious if an apples-to-apples comparison with Engel et al would be possible. As highlighted by Garcia and Randall (2011), the uncertainty on age of air may increase if you only sample it at a few locations and times, as opposed to globally. I suspect that model based estimate of uncertainty will increase markedly with limit sampling.

And finally, given the uncertainty associated with SF6 decay rates, I still worry that maybe our problem is being able to model SF6, as opposed, to being able to model age.
All this said, this was meant to be a minor suggestion. If the message is that models are still inconsistent, I would just highlight that there is a lot of uncertainty first, and then say that despite this, we cannot yet reconcile model trends with available observations.

Thank you for this constructive comment. We agree that currently, we are not doing a proper "apples-to-apples" comparison, and that this would be desirable. To achieve this there are a number of factors to be considered: 1) the sampling (addressed by Garcia &Randel (2011), but also for observational data by Ray et al. 2014), 2) the chemical tracer properties of SF6 and CO2, and 3) the non-linear time-evolution of the realistic tracers, and applying the same method to correct for that (as has been done by Fritsch et al, 2020). All those factors have been considered by various studies, but mostly only looking at those individually, and not in concert. Given that the CMIP models do mostly only supply zonal mean, monthly mean AoA values based on an idealized tracer, we cannot achieve this proper comparison here, but aim to tackle this in a future study. However, what can be done based on the CMIP data is testing for the effect of the sampling bias (though with the given the zonal mean, monthly mean data only). For this, we have applied the times and locations of the Engel et al 2009 measurements to the model AoA output. Figure R2 (Figure 6 in the revised paper) shows the resulting timeseries and trends. We have included the following paragraph on this on L184:
"We next examine the role of the limited sampling in the observational data on the detection of trends. Figure 6 shows timeseries and trends of AoA from the models subsampled at the locations and times of the Engel et al. (2009) measurements. Also included are observational estimates of the AoA and its uncertainty from observations, both from Engel et al. (2009) and from an updated version selecting different parameters in the derivation of AoA from the tracer measurements Fritsch et al. (2020). The latter present higher AoA values, larger uncertainties and smaller trends Fritsch et al. (2020). The trends are negative for every simulation, contrasting with the positive trends in the observational estimates (Fig. 6b). However, in this case the model trends are compatible with the observational trends from Fritsch et al (2020) in 14 out of the 32 simulations (43%), for which the model and observational trend error bars overlap. It is important to point out that the uncertainties in the subsampled model AoA trends are 5 times larger on average than those using all model data, going from a 10% to a 50% uncertainty on average (not shown). These large errorbars due to subsampling are responsible for the agreement within uncertainties with observations obtained for some of the model simulations. Finally we note that, if only the \chem{CO_2} measurement locations are considered, the model trends are compatible with zero in most of the simulations (not shown). This is because the early observations before 1985, which are key to get negative trends in the models, are based on SF_6 (Fig. 6a)"

[Figure]

Figure R2 (new Fig. 6 in the paper). (a) Timeseries of AoA in the models sampled as the observations. Observationally-derived estimates from Engel et al. (2009) and Fritsch et al. (2020) are also shown. (b) Trends in the above timeseries.

2) As noted by the authors, there term Brewer-Dobson Circulation has been used in many ways in the literature. As I feel this paper will become a very important reference point for the BDC, I would urge the authors to set a tone of best practices, and always refer to w* as the residual circulation (or the diabatic circulation / mean overturning circulation). An example where this would be helpful would be lines 323-4, where I think the authors mean to refer to changes in the residual circulation. Even though w* weakens in the southern hemisphere polar vortex (e.g. Figure 7), the age of air consistently decreases here. In the sense of tracer transport, then, the models are still suggesting an increase, even though w* has the opposite trend.

Note that I regret that I myself have used the terms loosely in the past! This meant as a minor suggestion.

Thank you for pointing this out. It is a crucial point, especially since we are trying to make the distinction and emphasize the role of mixing. We have revised all uses of BDC throughout the paper, including the one pointed out explicitly.

Minor suggestions by line number

2 consider "...in order to simulate surface climate variability and change."

changed to:  in order to simulate surface climate variability and change adequately

12 I would have thought the BDC describes the transport of *mass*, heat, and trace gases. The difference between the net transport of mass vs. trace gases is a nice way to highlight the role of isentropic mixing.

We agree and added 'mass'.

20 Consider deleting "which accounts for zonal asymmetries and" so that this reads " and two-way mixing, the irreversible tracer transport…"
My concern is that residual mean circulation depends fundamentally on eddies (in many regions, the "zonal mean" transport is in the opposite direction), and I wouldn't want a reader to think that eddies only matter for the mixing.

We agree that this can wrongly lead to think that eddies are irrelevant for the advective component and have removed that part.

27 Consider "transport diagnostic that quantifies the elapsed time"

changed

30 Linz et al. 2017 use AoA measurements to quantify the residual circulation. It might be fair to include a discussion of this paper here, or perhaps later on, in the discussion of observations. Linz et al. found that MIPAS SF6 age would imply huge problems with the reanalyses and models, or could reflect uncertainty in the lifetime of SF6.

We added a short discussion on this work in L132 (see Fig. R5 in this response to reviewers document):
"As a note, we applied a simple approximation of this method by computing the area-weighted age gradient on pressure levels (not shown) and found a relationship with the net upward mass flux for the 1pctCO2 runs (3 models), but not for the historical runs (4 models). We therefore cannot extract robust conclusions due to the limited number of models providing AoA output."

77 Given the small number of models, 66% might give an unfairly precise estimate of the uncertainty. There are only 8 models, so naively, you are saying 6/8 models must agree. But for the residual circulation, there are only 7, and AoA, only 5 at best. Perhaps you could say, we ask that at least 2/3rds of the model agree, which in practice meant 5 of 7 for diagnostics of the residual circulation, and 4 of 5 for the age of air.

We agree and have modified the sentence to: "We further ask for two thirds (2/3) of the models to agree (that is, 5 of 7 for the residual circulation and 4 of 5 for AoA)."

141 consider "which quantifies the influence of"

changed

165 It might be appropriate to also reference Linz et al. 2016, which makes this very explicit.

Citation added

Figure 5, the model AOA trends at 30 hPa in panel (b) are global mean trends, right? Is this a fair comparison with Engel et al. 2017, which I believe is based on measurements at 2 midlatitude NH locations? If nothing, else, the Engel trend should also be given with

significance: 0.15 +/- 0.18, or 0 to 6 %. And I would discuss the issue of sampling brought up by Garcia and Randal (2011), which I think is still relevant.

The model trends are averaged over 30-45N as stated in the figure caption, to match the range of latitudes of the observations. The issues with observational estimates comparison are now further discussed around the new Fig. 6, as described in the response to your first comment.

184 I think AoA converges faster not just because it has memory (integrating in time), but also because it integrates in space. The age at any point in the atmosphere depends not just on the local circulation, but on the integrated solution below. You could simply state "being an integrated quantity in both time and space."

Agreed, changed.

212 Weaker trends in the tropics relative to the high latitudes is consistent with the acceleration of the residual circulation. As suggested by Linz et al, 2016, increasing the residual circulation should reduce the gradient in age; hence a stronger reduction of midlatitude age. This result was first established by Neu et al. 1999 with the leaky pipe!

Thanks for pointing this out, we agree that some discussion on this feature would be beneficial. We have added the following sentences, based on this comment and a community comment by Petr Sacha: "The stronger AoA trends in the extratropics as compared to the tropics and subsequent reduction in age gradient are consistent with the overturning acceleration, as revealed by the leaky pipe model Neu et al. (1999). This feature has also been linked  to changes in mixing and the rise of the circulation Sacha et al. (2019)."

219 I worry that variability at 30 years here is Gibbs ringing. The 30 year box car average used to compute the trends will amplify any variability at this frequency relative to others.

We agree that this effect is very likely underlying the periodicity seen in the trends, but we have tested that the spread in the trends is still representative. We have added the following comment to warn the reader about the interpretation of this periodicity: "However, note that this apparent periodicity is an artifact of the 30 year period used to compute the trends related with the Gibbs ringing Gibbs (1898), as we have checked by changing the length of the period (not shown)."

Figure 10. I am curious if the kink in resolved wave forcing c. 7 hPa is due to issues with one model, or an artifact of the vertical resolution of the data set (such that it shows up in all the models).

This is observed in all the models, as seen in Fig. R3 below.

[Figure]

Figure R3. Vertical structure of contribution from different waves to the tropical upwelling trends in the various models.

254-6. I had to reread this a few times. I gather that the contribution of NOGW is uniformly small at this level, but the role of OGW is more uncertain; it plays a significant role in 4 models, and hardly any at all in 3.

Thanks for pointing this out. We have modified this sentence to clarify it, now it reads: "The contribution from OGW is more uncertain: it plays a significant role in 4 models (being the main forcing for GFDL-ESM4), but is negligible in the other 3 models."

265-8 This is an interesting result, which is consistent with the suggestion of Oberlander-Hayn et al. (2016) that much of the trend at this level can be understood as a lifting of the climatological overturning circulation. I appreciate that this paper is making a similar argument to al the studies listed at line 244-5, but I think the difference is the emphasis on why there is a trend. Downward control always makes one look above, while a lifting of the circulation points to the rise in the tropopause and the expansion of the troposphere in response to surface warming.

We agree that this is a sign of the upward shift of the circulation at the lower branch, but we are not sure that this applies also to trends in the deep branch, which are not directly associated with changes in the subtropical jets.

304 I'm not sure if I follow the argument here. The fact that the global vs. tropical sensitivity is different on interannual times scales (13a) suggests that it's naive to just consider the tropical SSTs.

The point is that the sensitivity is different but the correlations are similar for the global and tropical surface temperatures. On one hand, the fact that the correlation (thus $R^2$) is very similar for both global and tropical TS, confirms that the global temperatures are largely controlled by tropical TS. On the other hand, we argue that the sensitivity (deltaw*/deltaTS) is enhanced in the global mean because the globally averaged surface temperatures attenuate the main interannual variability signal which is linked to ENSO, such that deltaTS is small and the ratio (deltaw*/deltaTS) is larger. We have changed the order of the argument in the manuscript, as we think that it is clearer this way.

312-3 consider "Consistent with previous multi-model studies, there remains a clear disagreement ..." (just to avoid agreement /disagreement in the same sentence!) More importantly, please consider discussing this mismatch in more detail, as I do not think it is yet an apples-to-apples comparison. It might also be good to provide references to the previous modeling studies, and the observational studies as well. (I know they are provided elsewhere, but it's helpful for people who focus on the conclusions to read the paper quickly.)

Changed. We have added the following discussion:  "Consistent with previous multi-model studies, there remains a clear disagreement in the AoA trends between models and observations in the middle and upper stratosphere. On the other hand, there is very good agreement in the shallow branch trends. Even when the model's AoA is subsampled in space and time as the observations, the model trends remain negative in the middle stratosphere for all available models and ensemble members (Fig. 6). Nevertheless, for more than 40% of the available simulations the model and observed error bars overlap due to the very large observational uncertainty estimated in Fritsch et al. (2020)."

324 As noted in my general comment, might be good to say the residual circulation here, as opposed to BDC.

Changed.

**REVIEWER 3**

General comment:

This paper investigates the stratospheric Brewer-Dobson circulation in CMIP6 models in terms of residual circulation and mean age of air. Both the climatology and trends, for both past and future, are inter-compared. The results show quite some differences between the models regarding the climatology, but the paper states that all models are within the uncertainty range from observations. The model trends show a clear acceleration of the shallow branch but a less robust pattern of the deep branch, which is likely related to differences in wave forcing among models.
Overall, I regard this paper a very valuable model inter-comparison which is likely of high interest in the community. The paper is well written and the results are clearly presented. However, I think at some places the paper could still be improved to enhance its relevance, clarity and also some discussion could be placed into better context. Please note that the list of detailed comments below is clearly related to my strong interest in the subject and not to criticism of the paper. I rate my comments in this regard as minor and specific but would encourage the authors to elaborate a bit further on these, and hopw this would further improve the paper. After addressing the comments, I would strongly recommend publication.

Minor comments:

1. Comparison to CMIP5:
I can imagine that a key interest of readers when considering this paper concerns changes from CMIP5 to CMIP6 models. Several of these are briefly reported in the manuscript (e.g., L87, L266, L193). However, the paper could benefit significantly from a clearer presentation and discussion of these changes. In this spirit, I would suggest to include results from the previous CMIP5 project in several figures (regarding both climatology and trends), similar to what is done in Fig. 5 for CCMI models, for ease of comparison. Also, the various differences between CMIP5 and CMIP6 could be discussed together in a short subsection.

The updates from CMIP5 to CMIP6 are less relevant than the comparison to CCMI, since all models from the latter include a well-resolved stratosphere (with interactive chemistry), as well as all necessary TEM and AoA output, which is not the case for CMIP5. Therefore we have included CCMI in some figures to compare the results to the latter state of the art models. Still, we have explicitly compared our results to CMIP5 results where relevant throughout the paper, in particular referring to Hardiman et al. (2014), where the BDC was analyzed.

2. Comparison to observed trends:
This comment is related to the discussion of the comparison of mean age trends from models with those from balloon observations, mainly on p8/L150ff. I think this is a point of high interest to many readers. However, after reading the paper, at least to me, the question still remains: Do we really have a discrepancy between simulated and observed trends? The paper states at several places that there is "an inconsistency in BDC trends" (e.g., L4, L343), but also says that the recent results of Fritsch et al. (2020) and also potential sampling issues in the observational data (as argued by Garcia and Randel, 2011) could explain these differences. So my question is (related to a comment of another

Reviewer): How large is the remaining trend difference if one accounts for method and sampling uncertainties? If the proper sampling of model data is not possible, at least the discussion of these aspects could be clarified. And if the conclusions are not clear, I would avoid too strong related statements in the abstract and conclusions section.

In response to this comment as well as comment #1 from reviewer 2 (Ed Gerber), we have added a new figure examining the role of the limited sampling in observations. Please refer to the corresponding response to reviewer 2. There remain, however, uncertainties associated with the method of deriving AoA, which cannot be addressed in this study due to the lack of necessary output (see e.g. Fritsch et al. 2020).

3. Abstract:
I find the abstract somewhat unspecific and coming short in stating the main results of the paper. E.g., it is said that "CMIP6 results confirm the well-known inconsistency in BDC trends", or "paper reflects the current knowledge and main uncertainties" but it remains unclear what the "well-known inconsistency" or "current knowledge" are. I would recommend to avoid such unspecific terms but clearly state the results of the CMIP6 investigation regarding BDC climatology, trends, and forcing (even giving some numbers, e.g., MMM trend values).

We removed the last sentence of the abstract, as we agree that it was too vague. The rest of the results pointed out by the reviewer are already stated in the abstract, except regarding the forcing, on which we added a note: "The increasing $CO_2$ simulations feature a robust acceleration of the BDC but also reveal large uncertainties in the deep branch trends, possibly related to the parameterized gravity wave forcing."

4. Definition of deep branch:
In this paper, 1.5hPa is chosen as a characteristic level for the deep branch. The authors briefly say on p5/L99 that this level is substantially higher than used in most other studies (actually all studies I'm aware of). I don't really understand the reasoning given on the same page. Why should the fact that "upwelling minimizes at 1.5hPa" be a good reason for choosing this level as characteristic for the deep branch? As the 1.5hPa choice here is very different from other studies, it would be good to further clarify this argumentation. Also, I would recommend to add a discussion of previous studies on separating the deep from shallow branch and their criteria. Related questions I have are: Why do Lin and Fu (2013, Fig. 3) use 30hPa for the deep branch and find a seasonal cycle compared to the semi-annual cycle found here? In my view, also the Birner and Boenisch (2011) results would be more consistent with this choise and finding. Could it be that the 1.5hPa surface used here is indeed located above the actual BDC deep branch and the semi-annual cycle found here is actually related to the secondary circulation associated with the semi-annual oscillation and not the BDC?

The level of 1.5 hPa has been used before in the literature to describe the deep branch (e.g. Palmeiro et al. 2014). Note that Fu and Lin (2013) use the circulation above 30 hPa (not only at 30 hPa) as a metric for the deep branch. Rather than choosing a random level such as 10 hPa, we decide to have the criterion of maximum (not minimum) upwelling to define the level. Nevertheless, it is noted throughout the paper that the results are consistent at other levels (in particular regarding the wave forcing and the trend uncertainty).
Regarding the semiannual cycle, we have checked that it is linked to the fact that the NH and SH branches are of similar magnitude, in contrast with the lower stratosphere where the NH branch dominates. Some influence of the SAO will be embedded, but note that the SAO peaks at higher levels (~0.1 hPa, Smith et al. 2017).

We have modified the text to clarify this: "The semi-annual cycle in the upper stratosphere has been less studied. It is linked to the fact that the deep branch downwelling has similar magnitude in both hemispheres, in contrast with the lower stratosphere where the NH branch dominates. In addition, there is likely a contribution from the secondary circulation associated with the Semi-Annual Oscillation (e.g. Garcia et al. 1997, Young et al. 2011), although this peaks at higher levels (~0.1 hPa, Smith et al. 2017)."

5. Relation to surface warming:
It is stated that there is a "close connection between the BDC shallow branch and surface temp." (e.g., P16, L275), but I'm unsure how Fig. 12 proves that connection. If I understand the figure correctly, it just illustrates mass flux and surface temperature trends from the different models as ratios. If this is true, I don't see why this suggests causality. Wouldn't it be better to plot BDC trends vs. temperature trends for all models/simulations (e.g., scatter plots), to see whether those models with strongest surface warming also simulate strongest BDC trends?

We are not trying to prove the connection between the BDC shallow branch and the surface temperature in this paper. This connection including its underlying physical mechanism has already been well discussed in previous studies (e.g., Garcia and Randel 2008). The main focus here is  to assess the performance among CMIP6 models. We therefore think the comparison of the BDC sensitivity to surface warming is appropriate. Note that, while the surface warming drives a stronger BDC shallow branch, a model showing stronger surface warming does not necessarily simulate stronger BDC. This is because the BDC shallow branch strength is directly linked to the strength of the subtropical jets and hence the warming at the tropical upper troposphere (Garcia and Randel 2008, Lin and Fu 2013), but there is quite some inter-model spread of the ratio between the warming at the tropical upper troposphere and at the surface (Po-Chedley and Fu 2012). See Fig. 11 and Fig. 12 in Lin and Fu 2013 for example. That being said, panels b abd c in Fig. 4R (new Fig. 13) shows the scatter plot of the BDC trends versus the warming trends as the reviewer suggested. The intermodel correlations between the tropical surface warming trends are much larger with the shallow branch than with the deep branch. However, the relationship is less clear when considering only the historical experiments, where the surface warming trends are smaller and the larger inter-model variance that comes from the ratio between the upper tropospheric warming versus the surface warming dominates.

We have added the following text to describe the figure results: "Panels b and c in Fig. 13 further demonstrate the connection between (tropical) surface temperature and lower branch trends (Fig. 13b), as well as the absence of such connection for the deep branch (Fig. 13c). Specifically, there is a strong correlation across the model simulations between the shallow branch and surface temperature trends (panel b, $R^2 = 0.69$), and a much reduced correlation for the deep branch (panel c, $R^2 = 0.12$). However, note that the high correlation disappears if only the historical simulations are considered. This is because the connection occurs through the upper tropospheric warming, and there is substantial inter-model spread in the ratio of surface to upper tropospheric warming Po-Chedley and Fu (2012). This spread dominates in the historical simulations, which feature smaller warming trends as compared to the 1pctCO2. The results in panels b and c are qualitatively similar for the global surface temperature trends (not shown)."

[Figure]

Figure R4 (Fig. 6 in the paper): (a): Connections between long-term trends in mass flux in the shallow (70 hPa) and deep (1.5 hPa) branches and global and tropical mean surface temperature. The linear trends for the historical runs are calculated over the period 1960-2014 (triangles); for 1pctCO2 runs trends are calculated over years 1-150 (circles). Black errorbars show the MMM and one standard deviation among models. (b) and (c): Scatter plots of trends in mass flux at 70 hPa (b) and 1.5 hPa (c) versus tropical surface warming. The coefficient of determination R^2 across the models is shown in (b) and (c).

Specific comments:

P2, L41: I would explicitly state that the Fritsch et al. results also concern the age trends, e.g. add "...observational AoA and trend estimates... "

Added, thanks.

P6, L116: Regarding the comparison to observed age (Fig. 3) I would also state that a clear difference is the generally too weak extratropics-tropics age gradient in the models. Only CNRM and MRI models simulate subtropical gradients somewhat similar to observations, but these models show a high-bias in tropical age. In addition, I'm wondering how the age gradients here are related to upwelling differences (using the relation derived by Linz et al., 2016). At first glance, it seems to me that e.g. the GFDL model simulates a rather weak gradient but has a rather slow upwelling at 70hPa (Fig. 2) - this would actually be opposite as expected from the simple Linz et al. relation. Any ideas/comments?

Please see response to next comment.

P6, L120: I find this sentence about the Linz et al. paper at the end of this paragraph quiet confusing. Has the extratropics-tropics age gradient been used here?

We agree with the reviewer that this sentence was lacking context. We did check the connection between the two magnitudes and did not find clear correlation due to the limited number of models (and the fact that we are using pressure levels). We have added this sentence to clarify this point: "As a note, we applied a simple approximation of this method by computing the area-weighted age gradient on pressure levels (not shown) and found a relationship with the net upward mass flux for the 1pctCO2 runs (3 models), but not for the historical runs (4 models). We therefore cannot extract robust conclusions due to the limited number of models providing AoA output."

[Figure]

Figure R5. Scatter plot of climatology (left) and future changes (right) in AoA gradient (difference extratropics minus tropics, with midpoint at 35S/N) at 70 hPa versus tropical upwelling at 70 hPa. Simulations 1pctCO2 are shown in top panels, historical in bottom panels.

P7, L148: Maybe the sentence "reanalyses do not provide robust trend estimates" is too general. I understand that the result of Abalos et al. (2015) which is referred to here, is the fact that different w* estimates provide different trends for ERA-Interim. Aren't most estimates considered in that paper (8 out of 9) rather robust in their trends, at least in parts of the stratosphere? I would suggest to reformulate like "... for upwelling due to difficulties when calculating w* trends from reanalyses".

In Abalos et al. 2015 the sign of the trends in the lower stratosphere generally agrees in 8 out of 9 estimates, but there is a very large spread in the values of the trends. To better reflect this result, we have changed this to "We do not include reanalysis trend estimates for upwelling because these show a larger spread than the models and therefore do not help constrain the results."

Figure 4: What is the reason for the strange trend pattern in MMM 1998-2014 trends (Fig. 4f) in the tropics above about 10hPa? (A similar pattern occurs also in Fig. 7e).

This is likely linked to a residual QBO signal that might be present in the relatively short trends.

P10, L183: Why is there larger inter-annual variability in the deep branch than in the shallo branch? It would also be helpful to say what variability is represented in the considered simulations (here for the UKESM model), and how reliable this is. Can the

results based on UKESM simulations be generalized to other models, as the text here suggests?

To explain the larger uncertainty in the deep branch trends due to the larger role of internal variability, we point to the close coupling to surface temperature of the shallow branch, absent in the deep branch. We also mention a possible role of parameterized gravity waves. Regarding the generalization of the results, we exploit the large ensemble provided by UKESM in this figure, doing this with models providing only 3 members would not be as representative. Still, the results are expected to be consistent, since Fig. 5 shows a similar spread among ensemble members in UKESM and other models.

P11, L195: "A similar but more modest behavior is found in the CMIP6 MMM...". Actually, I can see this widening only for HadGem and WACCM.

In line with this comment as well as a similar comment from Rev. 1, we have rephrased this description: "A more modest widening is found in the CMIP6 MMM, limited to the NH, perhaps linked to a strengthening of the polar vortex in the MMM for the subset of CMIP6 models used in the present study (not shown). Nevertheless, we note that the trends in the polar vortex are highly model dependent, and for instance the two models that show a clear widening of the tropical pipe in the NH upper stratosphere (CESM-WACCM and HadGEM) features opposite-sign trends in the polar vortex."

Figure 7: Why is the upwelling region for the GISS model so broad? Is this realistic? Maybe add some comment on this.

The upwelling region in E2.2 is broader than the other models, at least above 10 hPa. Nevertheless, this does not compromise the fidelity of transport within the stratosphere, since the model does exhibit a realistic tropical tape recorder ascent and mean age (see Fig. 2 in Orbe et al. 2020). The model also exhibits realistic tracer gradients in the extratropics throughout the middle stratosphere in a range of species (Fig. 4 in Orbe et al. 2020). This suggests that this upwelling bias isn't having a substantial impact on the transport (at least in regions where we have satellite measurements).

P15, L255: Any idea why the GFDL model has that low contribution from resolved waves? (Similar for WACCM at 1.5hPa, L263).

We are not sure of the reasons for this and it would have to be examined in more detail. However, we note that the magnitude of the parameterized waves could play a role, given that the contributions from different waves can compensate and lead a robust net residual circulation across models (e.g. Cohen et al. 2014).

Figure 13: It would be good to add a measure of significance in Fig. 13 (and Fig. 12), e.g. the standard deviation from the regression as error bars.

Adding confidence intervals for each model in Fig. 13 would provide measures of the magnitudes of the trends relative to the internal variability. However, the purpose of this figure is to assess the multi-model performance, therefore we consider a more appropriate measure of the significance to show the inter-model spread relative to the multi-model mean signal, as shown in Figs. 12 and 13. For instance, consider the case where all of the models produce the same trend with weak amplitude. The trend is not significant when considering one model, but the trend is highly significant in the sense that all the models showing the same results. Moreover, adding error bars for each model would make the figure too busy.

Technical comments:

P2, L20: "Lagrangian"
Changed

P2, L26: "net strength in stratospheric tracer transport"
Changed to net stratospheric tracer transport

P3, L73: "significance in"
Added "in".

P11, L204: "substances"
Changed

P17, L293: delete "a" after "about"
Removed

**References**

Cohen, N. Y., Gerber, E. P., & Bühler, O. (2014). What Drives the Brewer–Dobson Circulation?, *Journal of the Atmospheric Sciences*, *71*(10), 3837-3855.

Fritsch, F., Garny, H., Engel, A., Bönisch, H., and Eichinger, R.: Sensitivity of age of air trends to the derivation method for non-linear increasing inert SF 6, Atmos. Chem. Phys, 20, 8709–8725, https://doi.org/10.5194/acp-20-8709-2020, https://doi.org/10.5194/acp-20-8709-2020, 2020.

Garcia, R. R., and W. J. Randel. "Acceleration of the Brewer-Dobson Circulation Due to Increase in Greenhouse Gases." *J.\ Atmos.\ Sci.* 65 (2008): 2731–39.

Gibbs, J. W.: Fourier's Series, Nature, https://doi.org/https://doi.org/10.1038/059200b0, 1898.

Hardiman, S. C., Butchart, N., and Calvo, N.: The morphology of the Brewer-Dobson circulation and its response to climate change in CMIP5 simulations, Quarterly Journal of the Royal Meteorological Society, 140, 1958–1965, https://doi.org/10.1002/qj.2258, 2014.

Kobayashi, C., and T. Iwasaki (2016), Brewer-Dobson circulation diagnosed from JRA-55, J. Geophys. Res. Atmos., 121, 1493–1510, doi:10.1002/2015JD023476.

Lin, P., and Q. Fu. "Changes in Various Branches of the Brewer–Dobson Circulation from an Ensemble of Chemistry Climate Models." *Journal of Geophysical Research: Atmospheres* 118, no. 1 (2013): 73–84.

Orbe, C., Rind, D., Jonas, J., Nazarenko, L., Faluvegi, G., Murray, L. T., et al. (2020). GISS Model E2.2: A climate model optimized for the middle atmosphere—2. Validation of large-scale transport and evaluation of climate response. *Journal of Geophysical Research: Atmospheres*, 125, e2020JD033151. https://doi.org/10.1029/2020JD033151

Neu, J. L. and Plumb, R. A.: Age of air in a "leaky pipe" model of stratospheric transport, Journal of Geophysical Research: Atmospheres, 104, 19 243–19 255, https://doi.org/10.1029/1999JD900251, http://doi.wiley.com/10.1029/1999JD900251, 1999.

Palmeiro, F. M., Calvo, N., and Garcia, R. R.: Future Changes in the Brewer–Dobson Circulation under Different Greenhouse Gas Concentrations in WACCM4, Journal of the Atmospheric Sciences, 71, 2962–2975, https://doi.org/10.1175/JAS-D-13-0289.1, http://journals.ametsoc.org/doi/abs/10.1175/JAS-D-13-0289.1, 2014.

Po-Chedley, S., and Q. Fu. "Discrepancies in Tropical Upper Tropospheric Warming between Atmospheric Circulation Models and Satellites." *Environ.\ Res.\ Lett.* 7 (2012).

Sacha, P., Eichinger, R., Garny, H., Pišoft, P., Dietmüller, S., De La Torre, L., Plummer, D. A., Jöckel, P., Morgenstern, O., Zeng, G., Butchart, N., and Añel, J. A.: Extratropical age of air trends and causative factors in climate projection simulations, Atmospheric Chemistry and Physics, 19, 7627–7647, https://doi.org/10.5194/acp-19-7627-2019, 2019.

Smith, A. K., Garcia, R. R., Moss, A. C., & Mitchell, N. J. (2017). The Semiannual Oscillation of the Tropical Zonal Wind in the Middle Atmosphere Derived from Satellite Geopotential Height Retrievals, *Journal of the Atmospheric Sciences*, *74*(8), 2413-2425. Retrieved May 28, 2021, from https://journals.ametsoc.org/view/journals/atsc/74/8/jas-d-17-0067.1.xml

---

## Author Response (AR2)

Dear Tim Dunkerton, we acknowledge your comments and provide responses below. We hope you find the paper ready for publication after addressing these comments.

Comments to the Author:
1. The relation between surface temperature and shallow BDC highlighted toward the end of the paper appears to be a simple form of "statistical attribution" which I describe, whenever encountered on Twitter, as "rubbish" because correlation is not causation. The extent to which statistical attribution has been stretched in public discussion is simply ludicrous, e.g., in the attempt to anticipate regional climate change from past statistical associations. The attribution of shallow BDC trend to surface temperature suggest common underlying dynamical mechanisms. This suggestion is implicit, but extremely important, as we attempt to construct a dynamical framework for climate change, particularly in zonally averaged metrics. I hope that authors will find suitable wording to be compatible with this comment.

We agree that correlation does not imply causation, and of course we base our conclusion on the robust dynamical mechanism connecting tropical surface temperature with the acceleration of the shallow branch. We thank the Editor for realizing that this mechanism was not explicitly stated, and we have now discussed it in lines 314-316: "This statistical relationship reflects an underlying dynamical mechanism: tropical surface warming leads to tropical upper tropospheric warming, which modifies meridional temperature gradients and thus wind shear (e.g. Garcia and Randel 2008), altering the wave propagation and dissipation conditions (Shepherd and McLandress 2011)."

2. For "direct" measurement of tropical upwelling, the classic tape recorder signal of tracers in HALOE provide strong constraints on the vertical profiles of upwelling, vertical diffusion, and lateral in-mixing; see Dunkerton (1997 JGR) for references and application to QBO wave driving. Later authors attempted to tease seasonal variation of upwelling also. These estimates nicely compliment age of air and provide, as it were, a local rather than time-integrated metric of BD transport.

We are aware that the tape recorder provides constraints on tropical upwelling (e.g. Mote et al. 1996). However, as you highlight, these measurements provide a measure of "effective" upwelling, which cannot be directly compared with w* from models, due to the role of horizontal and vertical mixing and diffusion. Unfortunately, tracer output, which would allow for more detailed comparisons of the net local tracer transport, is not generally available in these CMIP6 simulations.

3. As for the TEM statistics I'd like reassurance that daily or 4x daily fields were used for wave fluxes.

The TEM diagnostics used are monthly averages computed from daily averages (i.e., averages of each time step over a day).